# Robustness of Probabilistic Models to Low-Quality Data: A Multi-Perspective Analysis

**Liu Peng**[*]
Trustworthy and General AI Lab
School of Engineering, Westlake University
Hangzhou, China
LiuPeng_NGP@outlook.com

**Yaochu Jin**[†]
Trustworthy and General AI Lab
School of Engineering, Westlake University
Hangzhou, China
jinyaochu@westlake.edu.cn

## Abstract

This paper investigates a critical challenge in modern machine learning: how different probabilistic models withstand low-quality training data. Through a systematic, comparative investigation, we reveal a stark spectrum of robustness. Empirically, we find that autoregressive language models exhibit remarkable resilience against both token-level noise and structural corruption (for GPT-2, test NLL increases modestly from 2.87 to 3.59 despite 50% corruption). By sharp contrast, class-conditional diffusion models degrade catastrophically under identical noise levels (image-label consistency plummets by 56.81%), while image classifiers show a moderate vulnerability that diminishes with dataset scale. To explain these discrepancies, we analyze the results through a multi-perspective lens integrating information theory, PAC learning, and gradient dynamics. This framework identifies **what** informational properties drive robustness, **why** they are required for generalization, and **how** the optimization process achieves this resilience. These analyses suggest that robustness is heavily influenced by two key principles: the **richness of conditioning information**, which constrains the learning problem, and the **absolute information content** of the training data, which allows the signal from correct information to dominate statistical noise.

## 1 Introduction

Contemporary deep learning models are trained on increasingly vast datasets where the presence of low-quality data is inevitable (Radford et al., 2018; 2019; Brown et al., 2020; Podell et al., 2023b; Li et al., 2024). How models contend with such data, however, is far from uniform. Our systematic investigation reveals a stark divergence in robustness across modern probabilistic models: while autoregressive language models and large-scale classifiers are remarkably resilient to high levels of data corruption, class-conditional diffusion models exhibit catastrophic degradation under the same conditions.

This dramatic disparity, which synthesizes observations from privous work on discriminative model robustness (Rolnick et al., 2018) and generative model fragility (Na et al., 2023), motivates the central goal of this paper: to move beyond model-specific observations and uncover the fundamental principles governing this behavior. Why do some of the most powerful models in AI occupy opposite ends of the robustness spectrum?

To systematically probe this disparity, we conduct a suite of controlled experiments across these representative model families. Our methodology involves dynamically introducing quantifiable, random errors into the training data, allowing us to precisely control the level of corruption. This paradigm allows us study the effects of what we term **low-quality data**, which we define functionally as samples where the relationship between inputs, conditions, and target outputs has been corrupted in a way that is detrimental to the specific learning task.

---

[*]Liu is the family name.
[†]Corresponding Author

To answer this question, we adopt a multi-perspective analytical approach, integrating insights from information theory, PAC learning, and gradient dynamics. We hypothesize that the observed disparities can be explained by a coherent set of underlying factors. By integrating empirical findings with these theoretical viewpoints, we aim to provide foundational insights for understanding and predicting model robustness in real-world noisy environments.

The key contributions of this work are as follows:

- We conduct a systematic empirical investigation that validates and quantifies a stark divergence in robustness across autoregressive language models, class-conditional diffusion models, and image classifiers, providing controlled evidence for this critical phenomenon.

- We propose and apply a multi-perspective analytical framework that uses information theory, PAC learning, and gradient dynamics to explain **what** informational properties drive robustness, **why** they are formally required for generalization, and **how** the optimization process mechanistically achieves this resilience.

- Through this integrated approach, we identify two fundamental factors that govern model robustness: (1) the **richness of conditioning information** available to the model, and (2) the **absolute information content** of the training data.

## 2 RELATED WORK

The challenge of training on imperfect data is a central theme in machine learning, giving rise to a rich literature on noise robustness. For discriminative models, this is a well-established field; the surprising resilience of deep classifiers to label noise is well-documented (Rolnick et al., 2018; ZhangChiyuan et al., 2021), leading to an ecosystem of solutions, from noise-robust loss functions (Menon et al., 2019; Chen et al., 2020) to techniques for noise correction (Yi & Wu, 2019). More recently, attention has turned to the fragility of modern generative models. This has spurred a new wave of targeted, architectural fixes for issues like noisy labels in class-conditional diffusion models (Na et al., 2023) and corrupted contexts in language models (Gao et al., 2024). In parallel, empirical work has validated the principle that massive data volume can overwhelm supervision noise (Jia et al., 2021). While these approaches are vital, they focus on fixing individual vulnerabilities rather than explaining their origins.

To analyze such phenomena, our work draws upon several foundational theoretical frameworks. The **information-theoretic perspective** builds on the seminal work of Shannon (Shannon, 1948) and its application to neural networks, which frames learning as a process of preserving a useful signal from noisy inputs (Tishby & Zaslavsky, 2015). The **PAC learning framework** provides a formal link between a hypothesis class's complexity (e.g., its Vapnik-Chervonenkis dimension, which is typically matched to the task's complexity), the required volume of clean data, and the feasibility of generalization (Valiant, 1984). Finally, the **gradient-based perspective** offers a mechanistic view rooted in the extensive literature on stochastic gradient descent (SGD) dynamics, where factors such as the batch size and the nature of gradient noise are known to be crucial for optimization and stable learning (Keskar et al., 2017).

Our work departs from the prevailing focus on model-specific engineering to conduct a fundamental comparative investigation. Rather than chasing state-of-the-art performance on individual benchmarks, we aim to isolate the intrinsic properties that govern robustness across diverse model families. We are the first to systematically synthesize these distinct theoretical viewpoints to explain *why* a stark divergence in robustness exists between autoregressive, diffusion, and discriminative models. By integrating controlled experiments with this multi-perspective framework, we identify two core principles, **richness of conditioning information** and **absolute information content**, that provide a unified explanation for these disparate behaviors. A more comprehensive review of the literature is provided in Appendix F.

## 3 EXPERIMENTS

### 3.1 EXPERIMENTAL SETUP

Our experimental methodology is designed to precisely measure the impact of low-quality data under controlled conditions. We introduce noise at ratios ($r$) from 0.1 to 1.0 relative to the clean data volume, creating effective error rates ($e = r/(1 + r)$) up to 50.0%. We analyze the results using two complementary paradigms.

**Noise Generation Protocols.** To establish a foundational baseline for intrinsic robustness, our primary experiments employ unstructured, random noise. For text-based tasks, we corrupt target tokens by replacing them with tokens chosen uniformly at random from the entire vocabulary. For classification and class-conditional generation, labels are corrupted by replacement with a class chosen uniformly from the $C - 1$ incorrect alternatives. Full pseudocode is provided in Appendix G for reproducibility. While this stochastic corruption isolates the model's ability to extract signal from noise, we also investigate the impact of realistic, systematic errors through structured noise experiments in Section 3.3 and Appendix L.

**Primary Paradigm: Isolating Intrinsic Robustness.** For most of our experiments (autoregressive, diffusion, and classification models), our goal is to isolate the model's intrinsic tolerance to noise. To do this, we hold the amount of correct supervision constant by scaling the total training compute by $(1 + r)$. This design ensures that any performance degradation is a direct consequence of the added noise, not a lack of clean data. For stability in high-noise regimes, batch sizes were increased and iterations proportionally reduced to preserve this principle (see Appendix H).

**Secondary Paradigms: Fixed-Budget and Structured Noise.** While our primary experiments evaluate unstructured random noise under scaled training compute (increased total iterations or epochs), we introduce two crucial variations to broaden our analysis. First, to isolate the effect of noise under a strict computational constraint, we employ a **fixed-budget paradigm** (constant iterations) where clean tokens are directly swapped with unstructured, random noise (see Table 7 and Appendix J). Second, to move beyond random corruption, we introduce a **structured noise paradigm** for sequence-to-sequence experiments (Sec. 3.3). Here, the flawed target data is generated by an early-stage, partially trained version of the model. This provides a crucial test of our rich-context hypothesis under a more realistic, non-random error distribution that mimics machine-generated artifacts.

### 3.2 AUTOREGRESSIVE MODELS FOR TEXT GENERATION ARE ROBUST TO LOW-QUALITY DATA

To investigate the impact of incorrect data on the training of decoder-only transformer-based autoregressive models, we trained GPT-2 models (Radford et al., 2019) on the Open-WebText dataset (Gokaslan et al., 2019). OpenWebText is an open-source replication of the private Web-Text dataset originally used to train GPT-2 and comprises approximately 38 GB of text from 8,013,769 documents. The training set contains approximately 9 billion tokens, and the validation set contains approximately 4 million tokens.

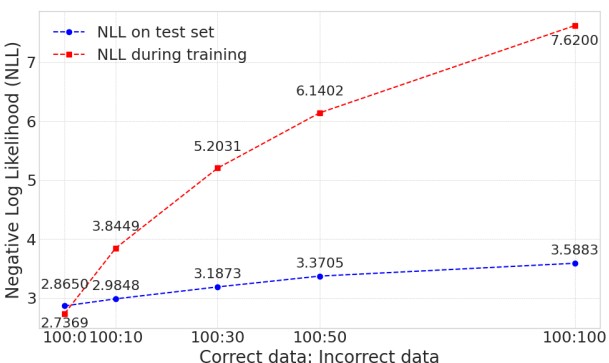

Figure 1: Impact of Increased Low-Quality Data on NLL. Results for 100:50 and 100:100 used increased batch sizes and proportionally reduced iterations to maintain training stability and equivalent correct sample exposure.

We trained 124M parameter GPT-2 models using the AdamW (Loshchilov & Hutter, 2017) optimizer. The baseline model was trained for 600,000 iterations. For experiments with added noise, the

total iterations were proportionally scaled to maintain constant exposure to the original clean data. The baseline batch size was maintained for most settings, but was selectively increased in high-noise regimes, a necessary intervention to overcome severe training instability. Full architectural and training configuration details are provided in Appendix H.

Figure 1 shows the negative log-likelihood (NLL) resulting from training language models with different ratios of additional incorrect data. The NLL on the test set represents the final evaluation after training, while the NLL on the (noisy) training set is reported from the end of the training process. Even when trained on data with a high error rate, language models can still achieve good performance on the test set. Notably, high ratios of additional incorrect data ($r = 0.5, r = 1.0$) introduced significant instability; training with baseline batch sizes failed to converge due to what we identify as overwhelming gradient noise. To counteract this, it was necessary to increase the batch size (doubling it for $r = 0.5$ and using a twelve-fold increase for $r = 1.0$) while proportionally reducing iterations to maintain equivalent exposure to correct samples. This necessary intervention provides direct empirical support for the gradient-averaging mechanism discussed in our analysis (Section 4.3). As the ratio of incorrect data increases, the NLL on the clean test set increases only slightly compared to the baseline (trained on correct data only), while the NLL on the noisy training set itself increases significantly with higher error rates. This phenomenon demonstrates that decoder-only transformer-based autoregressive models can learn effectively even in the presence of a substantial proportion of incorrect data.

To provide a complementary view under a **fixed computational budget**, we also analyzed performance where adding noisy data displaces clean data within a constant number of training steps. This analysis, detailed in Appendix I, reinforces our finding: even as the model attempts to fit the corrupted samples (leading to a high training NLL), its generalization to the clean data distribution remains largely intact (validation NLL increases only modestly). This further highlights the model's resilience.

### 3.3 THE PROTECTIVE EFFECT OF RICH CONDITIONING IN SEQUENCE-TO-SEQUENCE MODELS

We hypothesize that a model's robustness is profoundly influenced by the richness of its conditioning information, a principle we formally ground in information theory and PAC learning later in Section 4. Intuitively, a rich context severely constrains the possible outputs, reducing the complexity of the function the model must learn (as reflected by a lower effective VC dimension), thereby making it less susceptible to noise in the target. To test this prediction directly, we compare two sequence-to-sequence tasks with a vast informational disparity: WMT 2014 translation (Bojar et al., 2014) (sparse context, 99.9th percentile source length of 153 tokens) and CNN/DailyMail (Chen et al., 2016) summarization (rich context, 99.9th percentile source length of 2343 tokens).

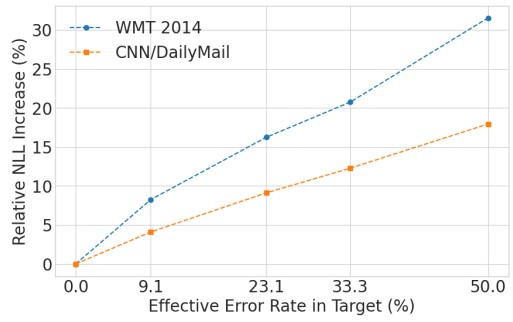

Figure 2: Relative NLL increase versus target noise. The model trained on CNN/DailyMail, with its information-rich conditioning input, is significantly more robust to target corruption than the WMT model, demonstrating how rich context constrains the learning problem.

To ensure a stringent test, we trained models from scratch and introduced **structured, non-random errors** into the target data using a "noisy teacher" paradigm. The results, shown in Figure 2, offer a clear empirical confirmation of our theoretical framework. At a 50% effective error rate, the NLL for the sparsely-conditioned WMT model degraded by 31.5%. By contrast, the richly-conditioned CNN/DailyMail model was far more resilient, with its NLL increasing by only 17.9%.

This finding provides strong evidence that robustness is not an inherent property of an architecture alone. Instead, it is heavily modulated by the information asymmetry between input and output.

When a model can draw upon a strong, constraining signal from a rich context, it can effectively average out and overcome substantial noise in a comparatively low-information target. The full results and experimental details are available in Appendix K.

### 3.4 CLASS-CONDITIONAL DIFFUSION MODELS ARE NOT ROBUST TO LOW-QUALITY DATA

To investigate the impact of substantial low-quality data on image generation, we trained class-conditional diffusion models and a classifier network separately on CIFAR-10 and CIFAR-100 (Krizhevsky, 2009). After training the diffusion model, we generated images by randomly selecting class labels as conditions. The pre-trained classifier then predicted labels for these generated images. We calculated an accuracy score by comparing these predicted labels with the conditioning labels used for generation.

We employ the EDM framework (Karras et al., 2022) for the diffusion model, using a U-Net architecture for the denoiser and a ResNet-18 model as the external classifier. Detailed hyperparameters for the diffusion process, network architectures, and training are available in Appendix H.

Algorithm 3 in Appendix G is used to generate incorrect labels. For a specific image, with a probability equal to the effective error rate $e$, its correct label was replaced by a new label randomly selected from the $C - 1$ alternative class labels.

Table 1: Ratio of Additional Incorrect Data and Corresponding Classification Accuracy of Generated Images (Consistency with Conditioning Labels) for Image Generation Tasks.

| Correct: Incorrect | CIFAR-10 Generation | CIFAR-100 Generation |
|---|---|---|
| 100: 0 | 94.082% | 65.236% |
| 100: 10 | 84.160% | 54.262% |
| 100: 30 | 69.864% | 38.882% |
| 100: 50 | 57.876% | 30.404% |
| 100: 100 | 40.630% | 16.996% |

The results in Table 1 show a substantial decrease in the accuracy of the generated images (consistency with the conditioning labels) as the proportion of incorrect training labels increases. For example, on the CIFAR-10 dataset, when 100% additional incorrect data is used (effective error rate $e = 0.5$, corresponding to the '100:100' condition), the accuracy drops from a baseline of 94.082% to 40.630%. For CIFAR-100, the impact is even more pronounced, with accuracy falling from 65.236% to 16.996% under the same conditions. Notably, the Fréchet Inception Distance (FID) scores for these generated images remained relatively stable across different levels of label incorrectness (see Appendix O for details). This suggests that the degradation in performance is primarily due to a weakened association between images and their conditioning labels, rather than a general decline in perceptual image quality.

### 3.5 ABSOLUTE INFORMATION CONTENT: CLASSIFIER ROBUSTNESS EMERGES AT SCALE

Table 2: Ratio of Increased Incorrect Data and Corresponding Accuracy for CIFAR Classification Tasks

| Correct:Incorrect | CIFAR-10 Classification | CIFAR-100 Classification |
|---|---|---|
| 100: 0 | 95.30% | 78.96% |
| 100: 10 | 95.11% | 77.33% |
| 100: 30 | 90.18% | 67.68% |
| 100: 50 | 89.19% | 63.71% |
| 100: 100 | 85.35% | 61.65% |

While autoregressive models demonstrated inherent robustness, the behavior of classifiers presents a more nuanced picture that powerfully highlights the role of dataset scale. On smaller datasets like CIFAR-10 and CIFAR-100, a ResNet-18 model trained from scratch exhibits moderate sensitivity to label noise, with performance degrading as corruption increases (Table 2). This establishes a baseline for moderately complex tasks with limited data.

Table 3: Ratio of Increased Incorrect Data and Corresponding Accuracy for ImageNet Classification Tasks

| Correct:Incorrect | ImageNet-10 | ImageNet-100 | ImageNet-1000 |
|---|---|---|---|
| 100: 0 | 62.302% | 64.520% | 73.784% |
| 100: 10 | 62.500% | 63.360% | 73.530% |
| 100: 30 | 58.929% | 57.560% | 73.646% |
| 100: 50 | 54.563% | 57.220% | 73.684% |
| 100: 100 | 50.794% | 45.920% | 74.778% |

To investigate whether a vast volume of correct signal can provide inherent resilience (a principle we formalize as **absolute information content** in Section 4.1.2), we scaled our experiments to ImageNet (Deng et al., 2009) using a ViT-Base model trained from scratch. The results in Table 3 are striking. While the ImageNet-10 and -100 subsets degrade similarly to CIFAR, the model trained on the full 1.28M-sample ImageNet-1000 dataset becomes almost impervious to label noise. Counter-intuitively, performance did not degrade but slightly improved, even when the training data contained 50% incorrect labels under the same setting, an effect we attribute to the additional training compute in our experimental design.

In high-noise regimes on the subsets, it was necessary to increase batch sizes to stabilize training—an empirical confirmation of the gradient-averaging mechanism we analyze in Section 4.3. This intervention, detailed in Appendix H, ensures a fair comparison. The extreme robustness on ImageNet-1000 thus provides compelling evidence that a sufficiently large volume of correct signal can dominate statistical noise. This robustness is further confirmed by our complementary **fixed-budget analysis** (see Appendix J), which isolates the effect from increased compute.

## 4 ANALYSIS

We analyze why autoregressive models and classification models can learn effectively despite substantial low-quality training data, while class-conditional diffusion models struggle under similar conditions. Our analysis is conducted from three complementary perspectives: information-theoretic, probably approximately correct (PAC), and gradient-based. This convergence analysis explains *what* informational properties drive robustness (information theory), *why* these properties are a formal requirement for generalization (PAC learning), and *how* the model mechanistically achieves this resilience (gradient dynamics). The analysis is built upon two fundamental principles. The first is the **richness of conditioning information**, which fundamentally governs a task's complexity. The second is the **absolute information content** of the data, which provides the learnable signal that can be mechanically extracted from noise via gradient aggregation.

### 4.1 INFORMATION-THEORETIC PERSPECTIVE

Information theory (Shannon, 1948), introduced to quantify information in communication, also offers a valuable lens for understanding machine learning as a process of information transfer to a model.

#### 4.1.1 RESIDUAL INFORMATION IN LOW-QUALITY DATA

To understand how models learn from corrupted data, we first quantify the amount of instructive signal that survives the introduction of noise. We measure this using **relative information loss**: the fraction of label uncertainty attributable to data corruption, normalized by the total entropy of the true labels. Let $\mathbf{y}$ be the true label and $\mathbf{x}$ be the observed (potentially corrupted) label from a set of $n$ classes. Assuming a uniform error model where an incorrect label is chosen randomly from the $n-1$ alternatives with probability $p_e$, the relative information loss is:

$$\frac{\text{information loss}}{H(\mathbf{y})} = \frac{-(1-p_e)\log_2(1-p_e) - p_e \log_2 p_e + p_e \log_2(n-1)}{\log_2 n} \tag{1}$$

This formulation (derived in Appendix Q) isolates the information-theoretic penalty of label noise itself. Analyzing this equation shows that for a large number of classes $n$, the loss increases

approximately linearly with the error rate $p_e$. Additionally, for a fixed error rate, the relative information loss decreases as $n$ grows.

These behaviors help explain the general performance degradation trends in our experiments (Section 3). However, the practical impact of $n$ is often coupled with other factors, such as the absolute data volume. The crucial insight from this analysis is that instructive information persists as long as the observed labels are not statistically independent of the true labels. This independence occurs at a single, precise point: when $p_e = (n-1)/n$. For error rates greater than this, the corrupted labels can paradoxically become informative again (e.g., $p_e = 1$ simply represents a perfectly inverted signal when $n = 2$). Our analysis and experiments operate in the realistic, information-degrading regime of $p_e \leq (n-1)/n$. Within this scope, a

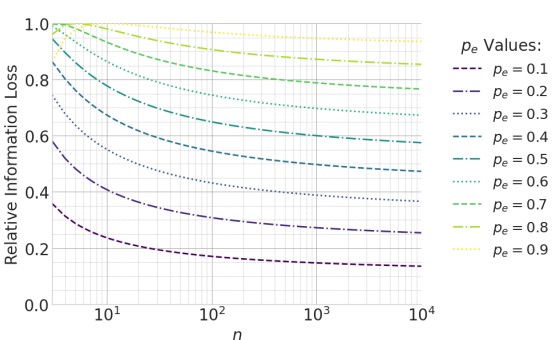

Figure 3: Behavior of Relative Information Loss with Varying $p_e$ and $n$.

residual signal always exists, allowing a model with sufficient capacity and data to extract meaningful patterns even from substantially noisy datasets. Figure 3 illustrates this behavior.

### 4.1.2 ABSOLUTE INFORMATION CONTENT AS THE PRIMARY DRIVER OF ROBUSTNESS

While the existence of residual information suggests that even corrupted data contains some useful signal, our analysis identifies the **absolute information content** of the training data as a principal driver of robustness. We define this as the total, aggregate quantity of information provided by the training data, comprising both the correct information for the input distribution $p(\mathbf{z})$ and the instructive information for the conditional distribution $p(\mathbf{y}|\mathbf{z})$, where $\mathbf{z}$ denotes the input.

It is crucial to distinguish what information a corrupted sample provides. An image with an incorrect label, for instance, still contributes to the model's understanding of the input distribution, $p(\mathbf{z})$, aiding the learning of robust visual features in a manner similar to unsupervised learning. Thus, while it provides zero instructive information for the supervised task itself, it still contributes to the absolute information content by enriching the model's representation of the input domain. Consequently, uncorrupted samples are essential as they provide the instructive component of the absolute information content that correctly guides the model to learn the input-output relationship.

Our experimental design directly investigates this principle. By scaling the total training duration by a factor of $(1+r)$, we ensure that across all experiments, the model is exposed to a constant quantity of this **correct instructive information**, holding the absolute information content relatively steady.

The remarkable robustness of the classifier trained on the full ImageNet-1000 dataset is a powerful illustration of this concept, especially when compared to smaller subsets like ImageNet-10 and ImageNet-100, which suffer much more severe performance degradation under identical noise rates. While the feature extractors learn from all processed images, the final classification is shaped by the immense absolute information content provided by the 1.28 million samples. This aggregate signal is so overwhelmingly strong that it provides a clear directive for the learning task, effectively allowing the model to average out and disregard the conflicting gradients from the noisy labels. This establishes that a sufficiently large absolute quantity of information, both in terms of raw input diversity and correct instructive signal, is a dominant factor in ensuring model robustness, explaining why massive datasets can often tolerate significant levels of noise.

### 4.1.3 THE ROLE OF RICHER CONDITIONING INFORMATION

We hypothesize that robustness is deeply influenced by the information asymmetry between the conditioning variables (inputs) and the target variables (outputs). Richer conditioning variables provide a more constrained and informative context, which can empower a model to overcome noise, particularly when that noise is in a comparatively information-sparse target.

This principle is demonstrated across our experiments. In our autoregressive language models, the conditioning context of previous tokens, $p(\text{next\_token}|\text{previous\_tokens})$, is information-rich compared to the single target token. Similarly, for image classifiers, the input image, $p(\text{label}|\text{image})$, contains vastly more information than the simple class label. Both of these model types proved robust when their information-sparse targets (the next token or the class label) were corrupted.

Conversely, our class-conditional diffusion models, $p(\text{image}|\text{class\_label})$, represent the opposite scenario. The conditioning variable (a single class label) is extremely information-sparse relative to the high-information target (a complete image). As predicted by our hypothesis, these models were highly fragile when this low-information conditioning signal was corrupted.

The sequence-to-sequence experiments in Section 3.3 provide an even more direct and compelling validation of this principle. We compared two tasks where the targets were corrupted: one with a short, less informative conditioning input (WMT 2014, with a 99.9th percentile source length of 153 tokens) and one with a long, information-rich conditioning input (CNN/DailyMail, 2343 tokens). The results were unambiguous: the model with the richer conditioning information (CNN/DailyMail) was significantly more robust, exhibiting only a 17.9% performance degradation compared to 31.5% for the model with the sparser input.

This demonstrates a clear pattern: models are vulnerable when low-information conditions are used to guide high-information outputs, but they can be remarkably robust when rich conditioning information provides a strong signal to overcome noise in simpler targets. This establishes the relative richness of the conditioning information as a key determinant of a model's resilience to low-quality data.

## 4.2 PROBABLY APPROXIMATELY CORRECT PERSPECTIVE

The Probably Approximately Correct (PAC) learning framework (Valiant, 1984) offers a theoretical lens through which we can understand the principles of richer conditioning information and absolute information content. PAC theory defines the sample complexity, $m$, as the minimum number of examples required to learn a concept with a low generalization error. For any concept class with a Vapnik-Chervonenkis (VC) dimension of $d$, this sample complexity $m$ is lower-bounded:

$$m \geq c_0 \left( \frac{1}{\epsilon} \log \frac{1}{\delta} + \frac{d}{\epsilon} \log \frac{1}{\epsilon} \right) \tag{2}$$

where $\epsilon$ and $\delta$ are the error and confidence parameters, and $c_0$ is a constant. (Kearns & Vazirani, 1994) This inequality reveals how both of our core robustness principles are grounded in learning theory.

First, the theoretical requirement for a minimum total number of samples, $m$, provides a foundation for our concept of **absolute information content**. As defined previously, this content comprises both structural knowledge and instructive signal. In deep neural networks, the global sample complexity $m$ is largely driven by the high capacity of the hypothesis class (e.g., the massive parameters of the shared feature extractor). Our ImageNet experiment clearly illustrates this: although subsets like ImageNet-10 and the full ImageNet-1000 share a similar number of samples *per class*, the full ImageNet-1000 dataset provides a vastly larger *aggregate* volume of clean examples across all classes. This massive total quantity ensures that, even under severe corruption, the absolute number of remaining clean examples across the entire dataset far exceeds the global sample complexity $m$ required to stabilize the shared learned knowledge. By contrast, the total number of uncorrupted examples in smaller subsets like ImageNet-10 is dangerously close to (or below) this critical threshold $m$, leaving no buffer to absorb the impact of noise.

Second, the VC dimension, $d$, which reflects the capacity of the hypothesis class the model belongs to, is tightly coupled with the principle of **richer conditioning information**. Strictly speaking, while $d$ is a property of the model architecture, the *required* value of $d$ to achieve a low approximation error is dictated by the complexity of the conditional distribution being modeled.

- **Richer Conditioning (e.g., Classification):** In tasks like $p(\text{label}|\text{image})$, the conditioning variable (image) is information-rich, while the target (label) is simple. The rich input severely constrains the possible outputs, simplifying the learning problem. This allows the task to be effectively solved by a hypothesis class with a lower effective VC dimension $d$.

- **Sparse Conditioning (e.g., Conditional Diffusion):** In tasks like $p(\text{image}|\text{label})$, the conditioning variable (label) is information-sparse, while the target (image) is extremely complex. The sparse input provides very little constraint, meaning the model must learn a far more complex mapping (e.g., high-dimensional score functions). This necessitates a hypothesis class with a much higher effective VC dimension $d$.

According to Inequality 2, a higher VC dimension $d$ demands a significantly larger number of samples $m$. Class-conditional diffusion models, with their sparse conditioning and consequently higher $d$, have an enormous requirement for absolute information content. This makes them exceptionally vulnerable to low-quality data, as noise rapidly depletes the effective number of clean samples below the critical threshold $m$ needed for successful learning.

Thus, the PAC framework converges with the information-theoretic perspective, identifying richer conditioning information (which lowers $d$) and sufficient absolute information content (which satisfies $m$) as the intertwined, principal drivers of a model's robustness to noisy data.

### 4.3 GRADIENT-BASED PERSPECTIVE

The training of modern neural networks via backpropagation (Rumelhart et al., 1986) provides a mechanistic explanation for *how* models achieve robustness to noisy data. This perspective highlights how aggregating samples amplifies the coherent signal from correct information while averaging out divergent noise from corrupted data, thereby leveraging the dataset's absolute information content.

Within any given training batch, the total gradient, $g_{total}$, can be decomposed into a coherent signal from correct samples and divergent noise from incorrect ones:

$$g_{total} = g_{correct\_signal} + \sum_j g_{noise\_component\_j} \qquad (3)$$

Here, $g_{correct\_signal}$ represents the consistent directional update from clean samples, guiding the model toward the true data manifold. By contrast, each $g_{noise\_component\_j}$ arises from a corrupted sample and points in a less predictable, often orthogonal, direction. To quantitatively validate this decomposition and the effect of sample aggregation, we analyzed per-example gradients at initialization across different data corruption ratios and batch sizes. The results are summarized in Table 4.

Table 4: Quantitative Analysis of Gradient Coherence. Clean gradients exhibit strong, **coherent positive alignment** (+0.52), while corrupted gradients are directionally random and orthogonal (similarity $\approx 0$). This disparity allows larger batch sizes to amplify the coherent signal relative to the noise, systematically improving the Signal-to-Noise Ratio.

| Metric | 25% Corruption | | 50% Corruption | |
|---|---|---|---|---|
| | **Batch Size = 4** | **Batch Size = 8** | **Batch Size = 4** | **Batch Size = 8** |
| *Directional Coherence (Mean Cosine Similarity)* | | | | |
| Clean vs. Clean | +0.52 | +0.52 | +0.52 | +0.52 |
| *(Min, Max)* | *(+0.00, +0.76)* | *(-0.00, +0.76)* | *(+0.00, +0.73)* | *(+0.00, +0.76)* |
| Corrupt vs. Corrupt | +0.001 | +0.001 | +0.001 | +0.001 |
| *(Min, Max)* | *(-0.02, +0.02)* | *(-0.02, +0.02)* | *(-0.01, +0.02)* | *(-0.01, +0.02)* |
| Clean vs. Corrupt | +0.001 | +0.001 | +0.001 | +0.001 |
| *(Min, Max)* | *(-0.03, +0.03)* | *(-0.03, +0.03)* | *(-0.03, +0.03)* | *(-0.03, +0.05)* |
| *Aggregated Signal Magnitude (Mean L2 Norm)* | | | | |
| Aggregated Clean Signal | 6.13 | 11.37 | 4.19 | 7.88 |
| Aggregated Noise Signal | 0.84 | 1.36 | 1.42 | 2.06 |
| **Signal-to-Noise Ratio** | **7.31x** | **8.34x** | **2.96x** | **3.83x** |

Table 4 provides direct empirical validation of our hypothesis. First, the coherence analysis confirms a fundamental disparity: gradients from clean data are consistently and strongly aligned (mean similarity +0.52), while gradients from corrupted data are directionally random, centered symmetrically around

zero and orthogonal to the clean signal. This holds true regardless of noise ratio or batch size. Second, and most critically, the table demonstrates the power of aggregation. For both 25% and 50% corruption levels, doubling the batch size causes the magnitude of the aggregated clean signal to nearly double, consistent with constructive accumulation. By contrast, the aggregated noise magnitude grows at a much slower rate, reflecting partial cancellation. Consequently, the signal-to-noise ratio systematically improves with a larger batch size in all scenarios. The full methodology and setup for this targeted gradient analysis are provided in Appendix M.

Beyond these per-step gradient dynamics, this fundamental mechanism of signal amplification has a direct, macroscopic consequence: it stabilizes the overall learning trajectory. To explicitly quantify this, we conducted a complementary experiment measuring inter-batch loss variance. As detailed in Appendix N, this dedicated supplementary analysis confirms that gradient aggregation drastically reduces loss variance. Together, these findings provide a complete mechanistic explanation for the empirical interventions required in our main experiments, where increasing the batch size was a necessary step to overcome severe data corruption.

Our specific intervention during the autoregressive model training (Section 3.2) vividly illustrates this principle in action. When baseline batch sizes led to instability in high-noise autoregressive model training, We increased the batch size up to twelve-fold to achieve convergence. As Figure 4(b) illustrates, this directly strengthens the cumulative $g_{correct\_signal}$ sufficiently to dominate the increased, but largely canceling, noise.

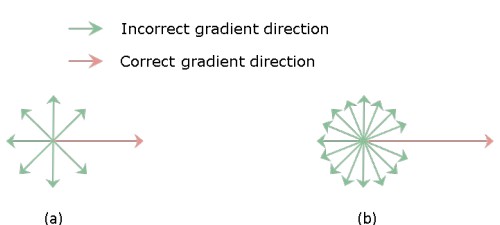

Figure 4: Interpretation of the gradient-based perspective. (a) Gradients from correct data are coherent, while those from incorrect data are divergent. (b) Aggregating samples in larger batches amplifies the correct signal relative to the noise.

Therefore, the gradient perspective confirms that aggregating samples is the crucial mechanism through which the statistical power of absolute information content is realized, enabling robust learning even with substantial low-quality data. This statistical averaging effect may be a fundamental reason why training large models often requires very large batch sizes (Yang et al., 2024; Touvron et al., 2023; Dubey et al., 2024; DeepSeek-AI, 2024).

## 5 CONCLUSION

This paper confronts a critical challenge in modern machine learning: the impact of low-quality data on probabilistic models. Our systematic investigation reveals a stark divergence in robustness across model families. We find that autoregressive models are remarkably resilient to significant data corruption, as are large-scale image classifiers. By sharp contrast, class-conditional diffusion models exhibit catastrophic degradation within our comparative analysis, pinpointing a critical vulnerability.

To explain these disparities, we analyze these results through a multi-perspective lens, integrating principles from information theory, PAC learning, and gradient dynamics to show *what* informational properties drive robustness, *why* they are formally required for generalization, and *how* this is mechanistically achieved. Our convergence analysis suggests that robustness in this context is heavily influenced by two key factors: the **richness of conditioning information**, which constrains the learning problem, and the **absolute information content** of the training data, which allows the aggregate signal from correct supervision to dominate the statistical noise from flawed examples. These principles move beyond model-specific observations to provide a more fundamental understanding of learning dynamics, offering crucial guidance for designing the next generation of reliable models intended for imperfect real-world data environments.

ACKNOWLEDGMENTS

Many thanks to Hengjie Yu, Yizhi Wang, and Peng Yue for their helpful discussions. This work is funded in part by an International Collaboration Fund for Creative Research of National Natural Science Foundation of China (NSFC ICFCRT) under the Grant no. W2441019.

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

# A  DISCUSSION

Decoder-only transformer-based autoregressive models for text generation are largely insensitive to low-quality data, which may partially explain their success (Radford et al., 2018; 2019; Brown et al., 2020; OpenAI et al., 2024). By contrast, class-conditional diffusion models exhibit greater sensitivity to data quality, suggesting that training such models requires a larger volume of high-quality data. This dichotomy is explained by our core finding: when rich conditioning information (e.g., a long text prefix) is available, models can overcome noise in a low-information target (the next token). When the conditioning is sparse (a single class label), the model is far more vulnerable. However, in the text-to-image task (Podell et al., 2023a), text conditions provide more information than categorical conditions, thereby reducing the amount of high-quality data needed for training.

A primary goal of our study was to establish this foundational understanding using a controlled, unstructured noise model, a necessary first step analogous to using a standardized test to measure a system's baseline capabilities. Our framework, however, also provides crucial foresight into the effects of more complex, structured noise, which presents a vital avenue for future work. Unlike the random errors studied here, which create diffuse gradient noise, structured noise introduces a coherent, competing learning signal. For example, a dataset where images of wolves are consistently mislabeled as "husky" would create a strong, incorrect gradient direction. Our gradient-based perspective predicts that this systematic error would be significantly more difficult to overcome, as the signal-averaging effect would be less effective against a persistent, biased signal, a prediction we confirm experimentally in Appendix L, where systematic mislabeling led to a catastrophic drop in classifier accuracy that was not observed with unstructured noise. In scoping our work, we distinguish between two types of structured noise that fall outside our primary research question. The first is correctable noise, such as the systematic wolf/husky error mentioned above, or the lexical misuse of "Complement" vs. "Compliment," which can often be solved contextually by other models. The second is noise from inherent ambiguity, such as the Trolley Problem, which lacks a single ground truth even for humans. By focusing on unstructured noise, our work addresses the more fundamental challenge of a model's intrinsic ability to find signal amidst stochastic corruption, a prerequisite for tackling these more complex scenarios.

The comparative validity of our theoretical framework is grounded in the clear disentanglement of model learning objectives. A class-conditional diffusion model, for instance, must solve two difficult problems simultaneously: learning the distribution for high-quality images and learning the correlation between those images and their labels. A classifier, by contrast, only needs to learn the correlation. We argue that our analysis is valid because our findings reveal a clear disentanglement of these two problems. For the class-conditional diffusion model, we show that label noise does not impact its ability to model image quality. Its capacity to learn the image manifold remains unimpaired, as evidenced by the stable FID scores detailed in Appendix O. The failure is catastrophic but also highly specific: it is isolated entirely to the label correlation, leading to a massive drop in image-label consistency as shown in Table 1. In stark contrast, the image classifier is largely robust. However, the sensitivity it does exhibit is clearly isolated to its correlation mechanism. This isolation is evident when comparing two factors: a moderate degradation in per-sample accuracy (Tables 2 and 3) against a near-perfect preservation of the marginal label distribution, which is quantitatively confirmed in Appendix P (KL Divergence < 0.0003). By isolating the correlation as the symmetric point of vulnerability, we can make a direct and insightful comparison, validating our analysis.

Our analysis intentionally focuses on models trained from scratch, rather than the dominant pre-training and fine-tuning paradigm, to establish a controlled, foundational understanding of robustness. This methodological choice is crucial for a clear interpretation of our results. Pre-trained models already possess a deep understanding of the world from their initial training, which acts as a powerful but confounding factor. By training from scratch, we remove this variable, allowing us to better isolate and understand the principles governing a model's robustness. Our findings then offer strong evidence that this robustness is shaped by the richness of conditioning information and the absolute information content of the data. Furthermore, the impact of fine-tuning can be transient: models exhibit a strong tendency to revert to their pre-trained behaviors, a phenomenon known as "elasticity" (Ji et al., 2025). Understanding how robustness is established in the initial training phase is therefore paramount, as this phase instills the core properties of the model. Our work provides this essential baseline, upon which future investigations into the more complex dynamics of fine-tuning with noisy data can be built.

The findings of this research contribute to a deeper understanding of how different probabilistic models handle imperfections in training data. This enhanced understanding can positively impact the machine learning community by enabling a more principled approach to data curation. One could potentially estimate the data quality requirements for a given model by considering the information asymmetry between its inputs and outputs. If the input is information-rich, data quality constraints can be relaxed; otherwise, a larger volume of high-quality data is necessary. As AI capabilities improve, driven in part by such foundational research, there is a potential for accelerated productivity across various sectors. However, it is also crucial to recognize that more powerful AI, stemming from a better grasp of its learning mechanisms, could also be misused for malicious purposes if not developed and deployed responsibly. Therefore, continued research on AI safety, ethics, and governance is paramount along with advancements in model capabilities.

## B    LIMITATIONS

The central goal of this paper is to provide a systematic and foundational analysis of how core model properties affect robustness. To achieve this, our experimental design primarily employs a simplified noise model: the dynamic introduction of **unstructured, random errors**. This approach ensures that the error rate is precisely quantifiable and reproducible, allowing us to isolate the effects of our core principles.

While our main experiments use unstructured noise to isolate core principles, we also validate our framework's predictive power on structured noise. In our sequence-to-sequence experiments (Section 3.3), we apply a form of **targeted, structured noise** to the output. Furthermore, our analysis of systematic label corruption in classifiers (Appendix L) confirms that our framework correctly predicts increased model fragility under such conditions. The primary limitation, therefore, is not the absence of structured noise analysis, but that our study does not systematically compare various forms of more complex, correlated noise (e.g., where errors depend on the input data). Exploring these scenarios is a crucial next step.

Additionally, our work has some other scope limitations. First, computational constraints precluded training diffusion models on very large-scale datasets. Second, we did not perform a direct analysis of the number of classes ($n$) as an independent variable, since its effects are inherently entangled with dataset size and model capacity. Third, we acknowledge that a direct comparison across tasks is challenging, as no unified metric exists for objectively scoring text, image, and classification models against one another. This is particularly relevant to our structured noise experiments comparing translation (WMT 2014) and summarization (CNN/DailyMail). While these are fundamentally different tasks, this was a deliberate choice to test our hypothesis under a clear disparity in context richness while controlling for model architecture and training configuration. We contend this is a more insightful proxy than intra-task comparisons, where a model trained on a long-context summarization task, for instance, might learn a trivial copying heuristic, or a long-context translation task could introduce output length as a new confounder. Our comparative insights are therefore derived from the starkly different relative degradation patterns each model exhibits against its own clean-data baseline. Finally, our experiments intentionally employ well-established and representative model architectures rather than the latest state-of-the-art systems. This choice is crucial for ensuring our findings are attributable to fundamental model properties, rather than confounding effects from specific, highly-tuned components of a particular SOTA model. The contribution of this work lies in analyzing principles of robustness, for which these architectures serve as clear and effective testbeds.

## C    REPRODUCIBILITY

Our work is designed to be fully reproducible. The complete source code, configuration files, and analysis scripts are publicly available at `https://github.com/LiuPeng-NGP/Robustness-Analysis`. For a full and transparent account of our methodology, we refer the reader to the detailed appendices, which document the precise training configurations (Appendix H), noise generation protocols (Appendix G), and other experimental specifics that form the basis of our findings.

## D    LLM USAGE STATEMENT

Large Language Models (LLMs) were utilized as an assistive tool in the preparation of this manuscript and its associated code. The LLM's role included: (1) improving the grammar and clarity of the text; (2) generating boilerplate code snippets; and (3) assisting in the articulation of authors arguments.

The fundamental scientific contributions, including the formulation of the key ideas, the experimental design, and the final interpretation of the results, are the original work of the human authors. The authors have critically reviewed, validated, and take full responsibility for all text and code presented.

# E PROBABILISTIC MODELS

Probabilistic models are widely used in machine learning to learn distributions from data. After training, the learned probabilistic model approximates the underlying data distribution. Probabilistic models can be broadly categorized into two types: generative models and discriminative models.

Generative models aim to learn the joint probability distribution $p_{data}(\mathbf{x}, \mathbf{y})$ or the data distribution $p_{data}(\mathbf{x})$. By learning this underlying distribution, generative models, such as those that model $p_{model}(\mathbf{x})$, can generate new data samples $\mathbf{x}$ that resemble those drawn from $p_{data}(\mathbf{x})$. Conditional generative models, which model $p_{model}(\mathbf{x} \mid \mathbf{y})$, generate data $\mathbf{x}$ based on specific inputs $\mathbf{y}$.

In contrast, discriminative models directly learn a decision boundary or the conditional probability $p_{model}(\mathbf{y} \mid \mathbf{x})$ of a label $\mathbf{y}$ given an input $\mathbf{x}$. They focus on predicting the label for a given input rather than modeling how the data itself is generated. Classification models are a prominent example of discriminative models, where the goal is to assign an input $\mathbf{x}$ to one of several predefined classes $\mathbf{y}$. Generative models generally require more sophisticated mechanisms to model complex distributions compared to discriminative models (Christopher Bishop, 2006).

Recently, generative models have achieved remarkable success across various domains, including text generation (OpenAI et al., 2024; Team et al., 2024; Dubey et al., 2024; Yang et al., 2024; DeepSeek-AI, 2024), image generation (Ho et al., 2020; Song et al., 2020a; Song & Ermon, 2019; Dhariwal & Nichol, 2021; Karras et al., 2022; Podell et al., 2023b), video generation (Kuaishou, 2024; OpenAI, 2024; Blattmann et al., 2023b; Singer et al., 2022b; Ho et al., 2022b), and audio generation (Borsos et al., 2023; Kreuk et al., 2022; Ziv et al., 2023; Kong et al., 2020b). Transformer-based autoregressive models (Vaswani et al., 2017) and diffusion models (Ho et al., 2020) have demonstrated exceptional capabilities in these areas.

## E.1 AUTOREGRESSIVE MODELS

Consider a sequence of random variables $\mathbf{x} = (x_1, \ldots, x_D)$, where each $x_i$ belongs to a defined domain. An autoregressive model decomposes the joint probability $p(\mathbf{x})$ as:

$$p(\mathbf{x}) = p(x_1) \prod_{d=2}^{D} p(x_d \mid x_{<d}). \tag{4}$$

Specifically, for text generation, autoregressive models generate the next token conditioned on previous tokens (Brown et al., 2020), while for image generation, they can generate the next pixel conditioned on previous pixels (Oord et al., 2016). Recurrent neural networks (Graves, 2014) (such as long short-term memory networks (Hochreiter & Schmidhuber, 1997)) and transformers (Vaswani et al., 2017) can be used as autoregressive models to generate data. Decoder-only transformer-based autoregressive models are currently prevalent for text generation (Radford et al., 2018; 2019; Brown et al., 2020; OpenAI et al., 2024) and audio generation (Borsos et al., 2023; Kreuk et al., 2022).

## E.2 DIFFUSION MODELS

Diffusion models (Sohl-Dickstein et al., 2015; Ho et al., 2020; Song & Ermon, 2019; Song et al., 2020b) have achieved remarkable success across various domains, including image generation (Chen et al., 2023b; Meng et al., 2021; Podell et al., 2023a), video generation (Ho et al., 2022a; Singer et al., 2022a; Blattmann et al., 2023a), audio generation (Liu et al., 2023; Yang et al., 2023; Kong et al., 2020a) and more (Wang et al., 2023b; Yu et al., 2024).

Diffusion models generate data by gradually denoising pure noise into meaningful data samples. The EDM formulation for diffusion models (Karras et al., 2022), proposed to elucidate the design space of diffusion models, is employed in this work to examine the influence of incorrect training data.

Assume $p_{data}(\mathbf{x})$ is the data distribution with standard deviation $\sigma_{data}$. Let $\sigma_0 = \sigma_{\max} > \sigma_1 > \cdots > \sigma_N = \sigma_{\min} \approx 0$ be a sequence of decreasing noise levels. We denote $p(\mathbf{x}; \sigma)$ as the marginal distribution of clean data samples from $p_{data}$ after being corrupted by i.i.d. Gaussian noise with standard deviation $\sigma$. Thus, $p(\mathbf{x}; \sigma_i)$ represents the distribution of data with noise level $\sigma_i$. In practice, the distribution at the maximum noise level, $p(\mathbf{x}; \sigma_{\max})$ (where $\sigma_{\max} = \sigma_0$), is indistinguishable from standard Gaussian noise. Diffusion models first sample random noise $\mathbf{x}_0 \sim \mathcal{N}(\mathbf{0}, \sigma_{\max}^2 \boldsymbol{I})$ and

then sequentially denoise it according to the noise levels. The result $\mathbf{x}_N$ thus aims to sample from the data distribution $p_{data}(\mathbf{x})$.

The probability flow ordinary differential equation (ODE) (Song et al., 2020b) is the deterministic counterpart of the stochastic differential equation (SDE), whose solutions describe a diffusion process. The probability flow ODE can continuously increase or reduce the noise level of the data depending on the direction of time:

$$d\mathbf{x} = -\dot{\sigma}(t)\sigma(t)\nabla_{\mathbf{x}} \log p(\mathbf{x}; \sigma(t))dt, \tag{5}$$

where the dot denotes the derivative with respect to time, and $\nabla_{\mathbf{x}} \log p(\mathbf{x}; \sigma)$ is the score function (Hyvärinen, 2005), which points in the direction of steepest ascent of the log-probability density $p(\mathbf{x}; \sigma)$.

The denoiser $D(\mathbf{x}; \sigma)$, which predicts clean data $\mathbf{y}$ from a noisy input $\mathbf{x} = \mathbf{y} + \mathbf{n}$ (where $\mathbf{y} \sim p_{data}$ and $\mathbf{n} \sim \mathcal{N}(\mathbf{0}, \sigma^2\boldsymbol{I})$), is trained by minimizing the following denoising score matching objective (Vincent, 2011):

$$\mathbb{E}_{\mathbf{y},\mathbf{n}}||D(\mathbf{y} + \mathbf{n}; \sigma) - \mathbf{y}||_2^2. \tag{6}$$

From the trained denoiser, the score function $\nabla_{\mathbf{x}} \log p(\mathbf{x}; \sigma)$ can be estimated as:

$$\nabla_{\mathbf{x}} \log p(\mathbf{x}; \sigma) = \frac{D(\mathbf{x}, \sigma) - \mathbf{x}}{\sigma^2}. \tag{7}$$

During inference, sampling is performed by substituting this estimated score function back into the probability flow ODE. Starting from initial Gaussian noise sampled at the maximum noise level, $\mathbf{x}_0 \sim \mathcal{N}(\mathbf{0}, \sigma_{\max}^2\boldsymbol{I})$, a numerical ODE solver (such as Euler's or Heun's method) is employed to integrate the trajectory backwards along the decreasing noise schedule from $\sigma_{\max}$ down to $\sigma_{\min}$. This deterministic integration process ultimately yields the final clean data sample $\mathbf{x}_N$ from the learned distribution $p_{data}$.

## F    RELATED WORK

The challenge of training models on imperfect data is a foundational issue in machine learning. Our work contributes by systematically analyzing how these imperfections affect modern probabilistic models, moving beyond model-specific fixes to uncover the general principles that govern robustness. This section situates our contribution by reviewing the literature on the data quality problem and the theoretical principles that inform our multi-perspective analysis.

### F.1    DATA QUALITY AND ROBUSTNESS IN DISCRIMINATIVE MODELS

The problem of data quality extends beyond simple label errors to include a range of imperfections like missing values and feature inaccuracies (Gong et al., 2023), all of which constitute a form of **information corruption**. Historically, the study of robustness to such corruption has centered on discriminative models. It is well-documented that deep classifiers can be surprisingly resilient to massive label noise (Rolnick et al., 2018), a finding that stands in tension with their ability to memorize random data (ZhangChiyuan et al., 2021). This observation has spurred the development of a rich ecosystem of methodological solutions, including techniques for noise correction (Yi & Wu, 2019) and the design of noise-robust loss functions (Menon et al., 2019; Chen et al., 2020). t

### F.2    EMERGING FRAGILITY IN GENERATIVE MODELS

While discriminative models have proven robust, the implications of information corruption for modern generative models present a distinct and more recent research frontier. These models are often tasked with learning highly complex, high-dimensional distributions, making them potentially more sensitive to noise.

Recent work has begun to document these vulnerabilities and propose targeted fixes. For instance, the sensitivity of class-conditional diffusion models has led to specialized solutions, such as transition-aware score matching (Na et al., 2023) and retrieval-augmented training (Chen et al., 2023a). Similarly, for language models, methods have been developed to mitigate noisy contexts during in-context learning (Gao et al., 2024) or fine-tuning (Wang et al., 2023a).

Our work shifts the focus from these model-specific solutions to a more fundamental question. Instead of asking *how* to fix a single model's sensitivity, we provide the first **systematic, comparative analysis** to explain *why* these starkly different robustness behaviors emerge. The fragility of diffusion models, in our view, is not a bug to be fixed but a key piece of evidence in this analysis.

Complementing this picture is the principle that massive data scale can often compensate for low data quality. The success of models trained on a billion noisy image-text pairs is a powerful demonstration of this effect (Jia et al., 2021). Our findings on large-scale classifiers align with this. We unify these seemingly disparate empirical observations under our proposed principles, suggesting that the **richness of conditioning information** and the **absolute information content** are the primary factors heavily influencing the observed spectrum of robustness.

### F.3    THEORETICAL FOUNDATIONS FOR ANALYZING ROBUSTNESS

Our analysis seeks to explain the observed disparities in robustness by synthesizing insights from three distinct but complementary theoretical viewpoints. While our convergent application of these perspectives is novel, each is grounded in established literature.

**The Information-Theoretic Perspective** frames learning as a process of extracting a useful signal from noisy data. Our analysis of robustness through the lenses of "richness of conditioning information" and "relative information loss" is a direct application of foundational concepts like entropy and mutual information (Shannon, 1948). This perspective allows us to quantify how much usable signal remains in corrupted data and explain why models with information-rich conditions (e.g., an image for a classifier, a long token history for an LM) are better equipped to handle noise in their targets than models with sparse conditions (e.g., a single class label for a diffusion model). This follows a tradition of analyzing neural networks through an information-theoretic lens (Tishby & Zaslavsky, 2015).

**The PAC Learning Perspective** provides a formal link between a task's complexity, the amount of data required, and the feasibility of generalization (Valiant, 1984). The theory establishes that more complex concept classes (i.e., those with a higher Vapnik-Chervonenkis dimension) require more clean samples to learn effectively. This principle helps to formally explain why inherently complex generative tasks, such as modeling the high-dimensional distribution of natural images, have a higher demand for information and are thus more sensitive to data corruption than comparatively simpler classification tasks.

**The Gradient-Based Perspective** offers a mechanistic explanation for how learning occurs amidst noise at the optimization level. The dynamics of stochastic gradient descent (SGD) are central to deep learning, and the inherent noise in the gradient estimation is known to have a regularizing effect (Zou et al., 2021). Our analysis builds on modern studies of these dynamics, which have highlighted the crucial roles of batch size in navigating the loss landscape (Keskar et al., 2017) and the anisotropic nature of gradient noise in escaping sharp minima (Zhu et al., 2019). This literature provides a firm basis for our argument that aggregating samples (e.g., through larger batches) strengthens the coherent "signal" from correct data against the chaotic "noise" from corrupted data, enabling effective and stable learning.

# G  ALGORITHMS

This section provides the algorithms used to calculate the error rate and generate low-quality data.

---

**Algorithm 1** Algorithm to Calculate Scaled Training Duration (assuming constant batch size) and Effective Error Rate $e$.

---

**Require:** $r$: The ratio of additional incorrect data relative to original correct data (e.g., r=1.0 means 100% additional incorrect data)

      $N_{orig}$: The number of original epochs or iterations

**Ensure:** $r \geq 0, N_{orig} > 0$

1.  $N_{new} \leftarrow N_{orig} \times (1 + r)$
2.  $e \leftarrow \frac{N_{orig} \times r}{N_{new}}$
3. **return** $N_{new}, e$

---

**Algorithm 2** Algorithm to Generate Incorrect Text Data

---

**Require:** $e$: The effective error rate (calculated as $r/(1 + r)$, see Algorithm 1)

      $V$: The size of the vocabulary

      $data$: The correct text data

      $B$: The batch size

      $L$: The sequence length

**Ensure:** $e \geq 0$

1.  $idx \leftarrow$ random_int$(0, \text{len}(data) - L, B)$ {Random starting indices}
2.  $X \leftarrow data[B, idx, idx + L]$ {Extract input sequences}
3.  $Y \leftarrow data[B, idx + 1, idx + L + 1]$ {Extract target sequence (shifted by 1)}
4.  $mask \leftarrow$ rand_like$(Y) < e$ {Create a mask for introducing errors}
5.  $rand\_vals \leftarrow$ randint_like$(Y, low = 0, high = V)$ {Generate random values for errors}
6.  $Y[mask] \leftarrow rand\_vals[mask]$ {Replace tokens where mask is true with random tokens}
7. **return** $X, Y$

---

**Algorithm 3** Algorithm to Generate Incorrect Image Labels

---

**Require:** $e$: The effective error rate (calculated as $r/(1 + r)$, see Algorithm 1)

      $y$: The true class label

      $C$: The number of classes

**Ensure:** $e \geq 0, C > 0$

1. **if** rand $< e$ **then**
2.   $possible\_labels \leftarrow$ list(range$(C)$)
3.   $possible\_labels$.remove$(y)$
4.   $incorrect\_label \leftarrow$ random_choice$(possible\_labels)$
5.   **return** $incorrect\_label$
6. **else**
7.   **return** $y$
8. **end if**

---

## H  TRAINING CONFIGURATION DETAILS

This section provides a summary of the batch sizes and training durations (iterations or epochs) used for the autoregressive language model experiments (Section 3.2) and the ImageNet classification experiments (Section 3.5). The configurations were designed to ensure that the total number of number of samples processed was scaled by a factor of $(1 + r)$ relative to the baseline (where $r$ is the ratio of added incorrect data), while the number of correct samples processed remained equivalent to the baseline.

### H.1  AUTOREGRESSIVE MODEL (GPT-2) TRAINING CONFIGURATION

The GPT-2 model architecture used (the 124M parameter version) consists of 12 transformer blocks. Each block sequentially applies Layer Normalization, Causal Attention, a second Layer Normalization, and a Multi-layer Perceptron (MLP). Each Causal Attention layer utilizes 12 heads. The model employs an embedding dimension of 768, a vocabulary size of 50,257, and has approximately 124 million parameters. Models were trained using the AdamW (Loshchilov & Hutter, 2017) optimizer with a weight decay of 0.1, $\beta_1 = 0.9$, $\beta_2 = 0.95$, and a maximum learning rate of $6 \times 10^{-4}$. The baseline model (0% added incorrect data) was trained for 600,000 iterations.

The baseline GPT-2 model ($r = 0$) was trained for $N_{\mathrm{orig}} = 600,000$ iterations with a baseline batch size of $B_{base} = 491,520$ tokens (12 samples/GPU $\times$ 1,024 sequence length $\times$ 5 gradient accumulation steps $\times$ 8 GPUs). For experiments with incorrect data, batch sizes and iterations were adjusted as detailed in Table 5.

Table 5: GPT-2 Training Configuration on OpenWebText. $N_{orig} = 600,000$ iterations. $B_{base}$ is the baseline batch size. Iterations are adjusted to maintain $(1 + r)$ scaling of total number of samples processed relative to baseline, keeping correct sample exposure constant.

| Correct:Incorrect ($r$) | Batch Size | Iterations |
|---|---|---|
| 100:0 ($r = 0$) | $1 \times B_{base}$ | $N_{orig}$ (600,000) |
| 100:10 ($r = 0.1$) | $1 \times B_{base}$ | $1.1 \times N_{orig}$ (660,000) |
| 100:30 ($r = 0.3$) | $1 \times B_{base}$ | $1.3 \times N_{orig}$ (780,000) |
| 100:50 ($r = 0.5$) | $2 \times B_{base}$ | $N_{orig} \times (1 + 0.5)/2$ (450,000) |
| 100:100 ($r = 1.0$) | $12 \times B_{base}$ | $N_{orig} \times (1 + 1.0)/12$ (100,000) |

### H.2  DIFFUSION MODEL AND CLASSIFIER CONFIGURATION

For the class-conditional diffusion models, we employ the EDM (Karras et al., 2022) framework with training settings: $\sigma_{data} = 0.5, p_{mean} = -1.2, p_{std} = 1.2$. For sampling, we use $\sigma_{\min} = 0.002, \sigma_{\max} = 80, \rho = 7,$ and $steps = 18$. The denoise network is a U-Net architecture (Ronneberger et al., 2015; Song et al., 2020b) with 15.7 million parameters. For training, we used a batch size of 128, a learning rate of 0.0001 with 200 warm-up epochs, and an exponential moving average decay rate of 0.9993 (Hunter, 1986). The classifier model is a ResNet-18 (He et al., 2016), trained for 200 epochs on CIFAR-10 and CIFAR-100, respectively.

### H.3  IMAGENET CLASSIFICATION (VIT-BASE) TRAINING CONFIGURATION

For the ImageNet experiments, we used the ViT-Base architecture (Dosovitskiy et al., 2020), which has 86M parameters. The baseline ViT-Base models ($r = 0$) for ImageNet classification tasks were trained for $N_0 = 300$ epochs with a baseline batch size of $B_0 = 128$ per GPU. For experiments with incorrect data, batch sizes and epochs were adjusted as detailed in Table 6.

Table 6: ViT-Base Training Configuration on ImageNet Subsets and Full ImageNet. $N_0 = 300$ epochs. $B_0 = 128$ (per GPU). Epochs are adjusted to maintain $(1 + r)$ scaling of total number of samples processed relative to baseline, keeping correct sample exposure constant.

| Dataset | Correct:Incorrect ($r$) | Batch Size (per GPU) | Epochs |
|---|---|---|---|
| **ImageNet-10** | | | |
| | 100:0 ($r = 0$) | $B_0$ (128) | $N_0$ (300) |
| | 100:10 ($r = 0.1$) | $B_0$ (128) | $1.1 \times N_0$ (330) |
| | 100:30 ($r = 0.3$) | $B_0$ (128) | $1.3 \times N_0$ (390) |
| | 100:50 ($r = 0.5$) | $B_0$ (128) | $1.5 \times N_0$ (450) |
| | 100:100 ($r = 1.0$) | $2 \times B_0$ (256) | $(1 + 1.0)/2 \times N_0$ (300) |
| **ImageNet-100** | | | |
| | 100:0 ($r = 0$) | $B_0$ (128) | $N_0$ (300) |
| | 100:10 ($r = 0.1$) | $B_0$ (128) | $1.1 \times N_0$ (330) |
| | 100:30 ($r = 0.3$) | $2 \times B_0$ (256) | $(1 + 0.3)/2 \times N_0$ (195) |
| | 100:50 ($r = 0.5$) | $2 \times B_0$ (256) | $(1 + 0.5)/2 \times N_0$ (225) |
| | 100:100 ($r = 1.0$) | $4 \times B_0$ (512) | $(1 + 1.0)/4 \times N_0$ (150) |
| **ImageNet-1000** | | | |
| | 100:0 ($r = 0$) | $B_0$ (128) | $N_0$ (300) |
| | 100:10 ($r = 0.1$) | $B_0$ (128) | $1.1 \times N_0$ (330) |
| | 100:30 ($r = 0.3$) | $B_0$ (128) | $1.3 \times N_0$ (390) |
| | 100:50 ($r = 0.5$) | $B_0$ (128) | $1.5 \times N_0$ (450) |
| | 100:100 ($r = 1.0$) | $B_0$ (128) | $2.0 \times N_0$ (600) |

## I  FIXED TRAINING COMPUTE FOR GPT-2

Table 7: Language Model NLL with Fixed Total Training Compute.

| Ratio of Clean to Noisy Data | Training NLL | Validation NLL |
|---|---|---|
| 100: 0 | 2.7369 | 2.8650 |
| 100:10 | 3.8744 | 2.9758 |
| 100:30 | 5.3622 | 3.1646 |
| 100:50 | 6.2423 | 3.3455 |
| 100:100 | 7.6048 | 3.6525 |

To further isolate the effect of noise from computational budget, we ran an additional analysis where we fixed the total training compute (i.e., total number of training steps) across all noise ratios. The results, presented in Table 7, provide further quantitative detail on this divergence. As the proportion of noisy data increases, the training NLL on the noisy data rises substantially, showing the model is attempting to fit the corrupted samples. By contrast, the validation NLL on clean data increases only modestly. This demonstrates the model's resilience; while its performance on the training distribution degrades, its generalization to the true, clean data distribution remains largely intact.

## J  FIXED TRAINING COMPUTE FOR IMAGENET CLASSIFIER

To provide a complementary view, we also conducted an analysis on ImageNet-1000 with a fixed computational budget, where adding noisy data means reducing the proportion of clean data seen per epoch. The results (Table 8) show that while training accuracy degrades significantly as the model attempts to fit the noisy labels, test accuracy remains remarkably stable, dropping by less than 3% even at a 50% error rate. This reinforces the finding that the model effectively learns from the true signal while averaging out the random noise.

Table 8: ImageNet-1000 Classification with a Fixed Total Training Budget.

| Ratio of Clean to Noisy Data | Training Accuracy | Test Accuracy |
|---|---|---|
| 100:0 | 94.244% | 73.784% |
| 100:10 | 92.728% | 72.992% |
| 100:30 | 87.162% | 72.302% |
| 100:50 | 80.533% | 71.854% |
| 100:100 | 66.870% | 71.093% |

# K DETAILED EXPERIMENTAL SETUP FOR SEQUENCE-TO-SEQUENCE ROBUSTNESS

This section provides a comprehensive overview of the experimental setup for the sequence-to-sequence robustness investigation presented in Section 3.3. Our objective was to rigorously compare the robustness of Transformer models in short-context versus long-context generation tasks when trained with structured target noise, while carefully controlling for confounding variables.

## K.1 DATASETS AND PREPROCESSING

- **Short-Context Task (Short-to-Short Generation):** We utilized the WMT 2014 English-German machine translation dataset. To ensure comparable data volume with the long-context task, the full WMT'14 training set was subsampled to 287,113 examples. The validation and test sets remained the original WMT'14 splits.

- **Long-Context Task (Long-to-Short Generation):** We used the CNN/DailyMail summarization dataset. Its training set naturally comprises 287,113 examples, providing an identical training data volume to the subsampled WMT'14.

- **Tokenization:** For fair comparison, both tasks employed separate, task-specific Byte-Level BPE tokenizers, each trained on its respective dataset's full text (source and target). A crucial control was setting the vocabulary size identically to 32,000 tokens for both WMT'14 and CNN/DailyMail tokenizers. This ensures equivalent embedding layer capacity across models.

- **Sequence Lengths:** We used a maximum sequence length of 256 for WMT 2014 (99.9th percentile) and 2048 for CNN/DailyMail. This 2048-token limit for CNN/DailyMail covers the vast majority of articles (95th pct: 1711; 99th pct: 2102; 99.9th pct: 2343), with target summaries capped at 256 tokens.

## K.2 MODEL ARCHITECTURE AND TRAINING

- **Model:** A standard Encoder-Decoder Transformer architecture was employed for both tasks. Models were trained entirely from scratch.

- **Hyperparameters:** Identical architectural hyperparameters were used across both tasks: 6 encoder layers, 6 decoder layers, 512 embedding dimension ($d_{model}$), 8 attention heads, 2048 feed-forward hidden dimension, and a dropout probability of 0.1.

- **Optimizer:** Adam optimizer with a learning rate of $1 \times 10^{-4}$ and gradient clipping at 1.0.

- **Training Duration:** All models were trained for 50k training steps, ensuring that models were exposed to a consistent number of total samples (clean + noisy) for each noise ratio, adhering to the "fixed-budget" paradigm for noise analysis.

## K.3 STRUCTURED NOISE GENERATION PROTOCOL

To introduce realistic, structured low-quality data into the target sequences, we employed a "noisy teacher" approach:

1. **Noisy Teacher Training:** For each task (WMT'14 and CNN/DailyMail), a clean Transformer model (the "Noisy Teacher") was trained on its respective *clean* dataset for an early,

fixed number of steps (e.g., 5,000 steps). This early-stage model is capable of generating text but produces outputs that are less coherent and accurate than a fully converged model, mimicking common forms of machine-generated errors.

2. **Noisy Target Generation:** The "clean_source" inputs from the training sets were fed into their respective "Noisy Teacher" models to generate "noisy_target" sequences. For machine translation, this produced poorly translated German sentences given English source. For summarization, this produced incomplete or inaccurate summaries given an article source.

3. **Mixed Training Datasets:** New training datasets were constructed where a specified percentage of the "clean_target" sequences were randomly replaced with these "noisy_target" sequences. Noise ratios of 0.1, 0.3, 0.5, and 1.0 were applied as other experiments, corresponding to effective error rates of 0.0909%, 0.2307%, 0.3333%, and 0.5%. The "clean_source" inputs always remained uncorrupted.

Table 9: Negative Log-Likelihood (NLL) on WMT 2014 and CNN/DailyMail with varying levels of structured target noise. These are the full results supporting the analysis in Section 3.3. Lower NLL is better.

| Ratio of Clean to Noisy Data | NLL (WMT 2014) | NLL (CNN/DailyMail) |
|---|---|---|
| 100:0 | 2.5488 | 3.3859 |
| 100:10 | 2.7591 | 3.5246 |
| 100:30 | 2.9625 | 3.6942 |
| 100:50 | 3.0777 | 3.8015 |
| 100:100 | 3.3525 | 3.9931 |

## L    ANALYSIS OF ROBUSTNESS TO STRUCTURED NOISE

A primary goal of our study was to establish a foundational understanding of robustness using a controlled, unstructured noise model. However, we acknowledge that real-world data imperfections are often structured. To test the predictions of our analytical framework under this more challenging condition, we conducted an additional set of experiments on CIFAR-10 and CIFAR-100 using a **structured noise** protocol.

The experimental setup, including the ResNet-18 model architecture and all training hyperparameters, was kept identical to the classification experiments in Section 3.5 to ensure a direct comparison. The sole modification was the noise generation mechanism. Instead of replacing a label with a randomly chosen incorrect class, we applied a systematic and consistent error: with a probability corresponding to the effective error rate, a true label $y$ was deterministically replaced with $(y + 1) \pmod{C}$, where $C$ is the total number of classes. This creates a coherent, competing signal, as all instances of a given class, when corrupted, are mislabeled as the same incorrect class.

The results, presented in Table 10, reveal a dramatically different picture of robustness compared to the unstructured noise scenario.

Table 10: Impact of Structured Label Noise on CIFAR Classification Accuracy. Unlike the diffuse gradients from random noise, the coherent incorrect signal from systematic mislabeling leads to a catastrophic performance decline, especially at high corruption rates.

| Correct:Incorrect | CIFAR-10 Accuracy | CIFAR-100 Accuracy |
|---|---|---|
| 100:0 | 94.17% | 75.54% |
| 100:10 | 94.15% | 75.83% |
| 100:30 | 91.28% | 74.05% |
| 100:50 | 87.99% | 62.44% |
| 100:100 | 40.72% | 33.39% |

While the model shows resilience at low levels of structured noise, its performance collapses at higher ratios. The contrast with unstructured noise is stark. For example, at a 50% effective error rate (the 100:100 condition) on CIFAR-10, accuracy plummeted to 40.72%, whereas the model maintained an accuracy of 85.35% under the same level of unstructured noise (Table 2). A similar catastrophic drop is observed for CIFAR-100, from 61.65% (unstructured) to 33.39% (structured).

This outcome provides strong validation for the gradient-based perspective detailed in Section 4.3. Unstructured, random noise generates divergent gradients ($\sum_j \boldsymbol{g}_{noise\_component\_j}$) that are directionally varied and can be effectively averaged out, allowing the coherent signal from correct data ($\boldsymbol{g}_{correct\_signal}$) to dominate. Structured noise, however, creates a coherent but incorrect gradient signal that systematically pulls the model parameters toward a wrong data manifold. This introduces a persistent, biased signal that cannot be canceled out through aggregation. The model is thus forced to learn a competing, incorrect hypothesis, leading to severe performance degradation. This experiment therefore confirms that our analytical framework not only explains robustness to random noise but also correctly predicts the increased fragility of models when faced with systematic errors.

## M    EXPERIMENTAL DETAILS FOR GRADIENT COHERENCE ANALYSIS

To quantitatively validate the claims made in our gradient-based perspective (Section 4.3), we conducted a dedicated experiment to analyze the directional properties and aggregate magnitudes of per-example gradients. This analysis is the source of the data presented in Table 4.

Our methodology mirrored the experimental context of our primary autoregressive model experiments (Section 3.2). We used a randomly initialized 124M parameter GPT-2 model, with the same architecture detailed in Appendix H, and sampled data from the OpenWebText dataset. We applied the same on-the-fly, unstructured noise protocol detailed in Section 3. The analysis focused on the word token embedding layer (`transformer.wte.weight`).

Per-example gradients were computed using the `torch.func.vmap` transform. We calculated two sets of metrics across 200 batches for each experimental condition: (1) **Directional Coherence**, measured by the pairwise cosine similarity between gradients from clean, corrupt, and mixed pairs and (2) **Aggregated Signal Magnitude**, the L2 norm of the sum of all clean gradients and, separately, all corrupt gradients within each batch. The Signal-to-Noise Ratio (SNR) was defined as the ratio of the mean L2 norm of the aggregated clean signal to the mean L2 norm of the aggregated noise signal.

The results, as shown in Table 4, provide strong empirical support for our theoretical claims. The analysis revealed a clear disparity in the directional coherence of the gradients, and we observed how the signal-to-noise ratio consistently improves with larger batch sizes.

## N    EXPERIMENTAL DETAILS FOR LOSS STABILITY ANALYSIS

To provide quantitative evidence for the gradient-averaging mechanism discussed in Section 4.3, we conducted a dedicated experiment to measure the stability of the training process under noisy conditions.

**Objective**    The goal of this experiment was not to measure generalization, but to quantify the consistency of the training signal itself. We hypothesized that while individual batches containing noisy data would produce chaotic gradients, aggregating samples into a larger "global batch" would yield a much more stable and consistent update direction. We use the inter-batch variance of the training loss as a direct proxy for the stability of the aggregated gradient.

**Methodology**    The experiment was conducted using the final model checkpoints from two of our GPT-2 training runs: the baseline model trained on 100% clean data, and the noisy model trained with a 50% effective error rate (the "100:100" condition).

The measurement process was as follows:

1. **Global Macro-Batch Definition:** A "global macro-batch" represents a single, large-scale gradient update step. Its size is defined as (*micro-batch size per GPU × gradient accumulation steps × number of GPUs*).
2. **Loss Calculation:** For each macro-batch, we processed multiple micro-batches of data drawn from the OpenWebText training set. The appropriate noise ratio (0% for the clean model, 50% for the noisy model) was applied on-the-fly. We recorded the training loss for each micro-batch on each GPU.
3. **Averaging:** The losses from all micro-batches within a single global macro-batch were averaged to produce a single, scalar loss value for that macro-batch.
4. **Statistical Analysis:** We repeated this process for 200 independent global macro-batches. The final reported metrics in Table 11 are the **mean** and **standard deviation** calculated over these 200 macro-batch loss values.

**Analysis**    These results provide direct quantitative evidence of how sample aggregation stabilizes learning. For a noisy model (50% corruption), the mean loss is significantly higher ($\approx 7.58$) compared to the clean baseline ($\approx 2.84$). It is vital to distinguish this high mean loss from the net direction of the parameter update. The elevated loss is an expected consequence of the objective accommodating the 50% corrupted labels, reflecting a necessary compromise in fitting the noisy data.

Table 11: Impact of Global Effective Batch Size on Gradient Signal Stability. The high mean loss for the noisy model is an expected consequence of fitting noise, whiles the sharp reduction in loss standard deviation with larger batches demonstrates increased training stability through gradient cancellation.

| Global Batch Size | Noisy Model (50% Corruption) | | Clean Model (Baseline) | |
|---|---|---|---|---|
| | Mean Loss | Std. Dev. $(\times 10^{-3})$ | Mean Loss | Std. Dev. $(\times 10^{-3})$ |
| 480 | 7.5806 | 9.45 | 2.8366 | 17.50 |
| 960 | 7.5811 | 7.08 | 2.8377 | 13.17 |
| 1920 | 7.5809 | 4.58 | 2.8373 | 8.85 |
| 3840 | 7.5805 | 3.61 | 2.8369 | 6.43 |
| 7680 | 7.5807 | 2.40 | 2.8375 | 4.44 |

However, the stability of the learning process is revealed by the loss variance. At first glance, the noisy model in Table 11 appears more stable, exhibiting a lower loss standard deviation than the clean baseline. This is a statistical artifact: the consistently high, low-variance loss from corrupted random targets statistically dampens the natural, higher variance from the clean data. The crucial insight, therefore, comes not from the absolute variance but from its trend. As the global effective batch size scales from 480 to 7680, the inter-batch standard deviation of the loss, a direct proxy for gradient stability, is reduced by approximately 75% in both noisy and clean scenarios. This sharp reduction signifies that the aggregated gradient provides a stable and consistent update direction. Although the noisy samples dampen the overall gradient magnitude, the coherent signal from the correct samples remains dominant after the divergent noisy gradients partially cancel each other out. This enables a reliable optimization trajectory that, over many steps, allows the model to learn the true data distribution, explaining its strong generalization despite the high training loss.

## O  FID FOR IMAGE GENERATION

Table 12 shows the FID calculated for the diffusion model in Section 3.4. FID was calculated using 50,000 generated images and the original dataset images, employing the "pytorch-fid" package (Seitzer, 2020). Even with an increased proportion of incorrect conditioning labels in training, the FID scores remained largely unchanged. The relatively stable FID scores across different levels of incorrect data suggest that the observed drop in classification accuracy for class-conditional diffusion models is primarily due to a mismatch between generated images and their conditioning class labels, rather than a degradation in the perceptual quality of the generated images themselves.

Table 12: Ratio of Increased Incorrect Data and Corresponding FID for Image Generation.

| Correct: Incorrect | CIFAR-10 Generation | CIFAR-100 Generation |
|---|---|---|
| 100: 0 | 3.49 | 5.38 |
| 100: 10 | 3.68 | 5.70 |
| 100: 30 | 3.66 | 6.09 |
| 100: 50 | 3.66 | 6.12 |
| 100: 100 | 3.62 | 6.28 |

## P  ANALYSIS OF LEARNED LABEL DISTRIBUTIONS IN CLASSIFIERS

This section details a supplementary experiment conducted to quantitatively validate the claim made in our Discussion (Section A). The goal is to demonstrate that the image classifier successfully learns the marginal label distribution, even when its per-sample conditional accuracy is degraded by label noise. This provides the empirical basis for our argument that the classifier's sensitivity to noise is isolated to its conditional guidance (correlation), not its understanding of the output space's structure.

For this dedicated analysis, we replicated the training process for the CIFAR-10 classification experiments presented in Section 3.5. This involved retraining the ResNet-18 models under identical architectural and hyperparameter configurations for each noise level. While minor variations exist due to training stochasticity, the final test accuracies of these replicated models are consistent with those reported in Table 2, confirming that they exhibit the same fundamental robustness characteristics.

The evaluation process was as follows:

- Each newly trained model (corresponding to effective error rates of 0%, 9.1%, 23.1%, 33.3%, and 50%) was run on the full, clean CIFAR-10 test set (10,000 images).
- We collected the complete set of 10,000 predicted labels generated by each model.
- We then compared the statistical distribution of these predicted labels against the true, uniform distribution of the test set labels (1,000 samples per each of the 10 classes).

To quantify the similarity between the predicted and true label distributions, we employed two standard metrics:

- **Kullback-Leibler (KL) Divergence:** Measures how one probability distribution diverges from a second. A KL Divergence value close to zero indicates that the two distributions are nearly identical.
- **Total Variation Distance (TVD):** Measures the total difference between two probability distributions. A TVD value close to zero also signifies high similarity.

The results, summarized in Table 13, reveal a stark contrast between the model's conditional performance and its grasp of the marginal label distribution.

As shown in the table, while the model's test accuracy, which is a measure of its per-sample conditional mapping ability, $p(\text{label}|\text{image})$, degrades under high noise ratios, the KL Divergence and TVD remain extremely low and stable across all conditions. A KL Divergence of $\approx 0.0002$

Table 13: Impact of Label Noise on Conditional Accuracy vs. Learned Marginal Distribution (Replicated CIFAR-10 Runs). While per-sample accuracy degrades, the KL Divergence and TVD remain exceptionally low, indicating the model consistently learns the true underlying label distribution.

| Correct:Incorrect Ratio | Effective Error Rate | Test Accuracy (Conditional) | KL Divergence (Marginal) | Total Variation Dist. (Marginal) |
|---|---|---|---|---|
| 100:0 | 0.0% | 93.85% | 0.000228 | 0.0088 |
| 100:10 | 9.1% | 94.08% | 0.000241 | 0.0089 |
| 100:30 | 23.1% | 92.11% | 0.000068 | 0.0045 |
| 100:50 | 33.3% | 87.96% | 0.000152 | 0.0076 |
| 100:100 | 50.0% | 86.28% | 0.000155 | 0.0077 |

signifies that the distribution of the model's 10,000 predictions on the test set is statistically almost indistinguishable from the true uniform distribution.

This provides powerful empirical support for the argument presented in our Discussion. This demonstrates that the classifier successfully learns the correct marginal distribution of the output space. Even when its conditional, per-sample predictions are less accurate, its aggregate predictions reproduce the true statistical frequencies of the test set. The performance degradation is therefore isolated to the conditional guidance mechanism (the correlation). This finding is crucial, as it validates our comparison with the class-conditional diffusion model, which exhibits an analogous failure mode: its knowledge of the output structure (image quality) is preserved, while its conditional guidance (label correlation) collapses.

## Q  RELATIVE INFORMATION LOSS

Let $\mathbf{y}$ represent the true label and $\mathbf{x}$ the observed label provided to the model during training (which may be corrupted from $\mathbf{y}$ with probability $p_e$). Let $n$ be the number of label classes, and let $p_e$ be the error rate, which is the probability that any given label is incorrect. Additionally, assume the classes follow a uniform distribution, such that $p(i) = \frac{1}{n}$.

The entropy of the true labels under a uniform distribution is:

$$H(\mathbf{y}) = -\sum_{i=1}^{n} p(i) \log_2 p(i) = -\sum_{i=1}^{n} \left(\frac{1}{n}\right) \log_2 \left(\frac{1}{n}\right) = \log_2 n \tag{8}$$

If the labels are mislabeled with an error rate $p_e$, the observed labels are correct with a probability of $1 - p_e$. Furthermore, we assume the incorrect classes follow a uniform error distribution, meaning each piece of data can be mislabeled as any of the $n - 1$ incorrect labels with probability $\frac{p_e}{n-1}$. The conditional entropy is then:

$$H(\mathbf{y} \mid \mathbf{x}) = -\sum_{i=1}^{n} p(i) \left[ (1 - p_e) \log_2(1 - p_e) + \sum_{j \neq i} \frac{p_e}{n-1} \log_2 \left(\frac{p_e}{n-1}\right) \right] \tag{9}$$

Since $p(i) = \frac{1}{n}$ for all i:

$$H(\mathbf{y} \mid \mathbf{x}) = -\sum_{i=1}^{n} p(i) \left[ (1 - p_e) \log_2(1 - p_e) + \sum_{j \neq i} \frac{p_e}{n-1} \log_2 \left(\frac{p_e}{n-1}\right) \right] \tag{10}$$

$$= -\frac{1}{n} \sum_{i=1}^{n} \left[ (1 - p_e) \log_2(1 - p_e) + (n - 1)\frac{p_e}{n-1} \log_2 \left(\frac{p_e}{n-1}\right) \right] \tag{11}$$

$$= -\left[ (1 - p_e) \log_2(1 - p_e) + (n - 1)\frac{p_e}{n-1} \log_2 \left(\frac{p_e}{n-1}\right) \right] \tag{12}$$

$$= -(1 - p_e) \log_2(1 - p_e) - p_e \log_2 p_e + p_e \log_2(n - 1) \tag{13}$$

If we use the difference between the entropy of the true labels and the mutual information to represent information loss, then:

$$\text{information loss} = H(\mathbf{y}) - I(\mathbf{x}; \mathbf{y}) \tag{14}$$

$$= H(\mathbf{y}) - (H(\mathbf{y}) - H(\mathbf{y} \mid \mathbf{x})) \tag{15}$$

$$= H(\mathbf{y} \mid \mathbf{x}) \tag{16}$$

The ratio of the information loss to the total entropy, which we define as the relative information loss, becomes:

$$\frac{\text{information loss}}{H(\mathbf{y})} = \frac{-(1 - p_e) \log_2(1 - p_e) - p_e \log_2 p_e + p_e \log_2(n - 1)}{\log_2 n} \tag{17}$$

For $\mathbf{x}$ to be independent of $\mathbf{y}$, the observed label $\mathbf{x}$ must provide no information about $\mathbf{y}$. This occurs when the probability of "observing the correct class" is equal to the probability of "observing any specific incorrect class." Under uniform label noise, this reduces to:

$$P(\mathbf{x} = i \mid \mathbf{y} = i) = P(\mathbf{x} = i \mid \mathbf{y} = j) \quad \forall j \neq i. \tag{18}$$

Substituting the noise model probabilities:

$$1 - p_e = \frac{p_e}{n-1} \tag{19}$$

Solving for $p_e$:

$$(n-1)(1-p_e) = p_e \implies n-1-(n-1)p_e = p_e \implies n-1 = np_e, \tag{20}$$

$$p_e = \frac{n-1}{n} \tag{21}$$

Thus, when $p_e = \frac{n-1}{n}$, the observed labels $\mathbf{x}$ contain no information about the true labels $\mathbf{y}$, and the relative information loss reaches its maximum value of $1$.

