# OpenReview forum: "Robustness of Probabilistic Models to Low-Quality Data: A Multi-Perspective Analysis"
_ICLR.cc/2026/Conference — ICLR 2026 Poster_

### Official Review · Reviewer_rhZG · 2025-10-29

**Soundness:** 4
**Presentation:** 4
**Contribution:** 4
**Rating:** 6
**Confidence:** 3

**Summary:**

This paper presents a systematic experimental analysis of the sensitivity of mainstream probabilistic models—specifically autoregressive, diffusion, and classifier models—to low-quality data. The authors identify two primary factors governing robustness: the "richness of conditioning information" and the "absolute information content" of the training data. The paper further provides theoretical grounding for these findings from the perspectives of information theory, PAC learning, and gradient dynamics.

**Strengths:**

1. The paper provides a systematic and logically self-consistent experimental investigation that effectively demonstrates the varying tolerances of autoregressive models, diffusion models, and image classifiers to low-quality data.
2. The two key factors proposed as governing robustness—the "richness of conditioning information" and the "absolute information content"—are insightful and provide a strong conceptual framework for understanding the observed phenomena.
3. The explanatory analysis accompanying the experimental results is thorough and convincing, particularly the multi-perspective theoretical support that unifies the empirical findings.
4. The paper's technical details are clearly presented, and the experimental setup appears to be highly reproducible.

**Weaknesses:**

1. The study's focus on training models "from scratch" (as noted in Sec 3.5) overlooks the dominant "pre-training and fine-tuning" paradigm used in modern practice. The robustness of a large model pre-trained on massive datasets may differ significantly when fine-tuned on low-quality data. This omission limits the direct applicability of the paper's findings to many real-world scenarios.
2. The analysis relies almost exclusively on unstructured, random noise (as acknowledged in Appendix B). This is a significant limitation, as real-world data noise is often structured or correlated (e.g., systematic mislabeling). It would substantially strengthen the claims to include even a simple experiment with structured noise. For instance, in the classification task, could the authors investigate a scenario where noise is not uniformly random but fixed to specific incorrect labels (e.g., all instances are mislabeled as class 'B')?

**Questions:**

1. Following on from weakness #2, the paper's conclusions on robustness, especially the explanation rooted in gradient averaging (Sec 4.3), are derived from experiments using unstructured random noise. How well would these principles generalize to real-world, structured noise patterns where the "noise" gradients are coherent, not random, and thus would not average out?
2. The experimental design in Sec. 3.1 attempts to isolate noise effects by fixing the “equivalent correct sample exposure,”but this approach changes both batch size and iteration count, introducing confounding factors. A control experiment with fixed batch size and varying noise ratio would better isolate the true impact of gradient averaging from training configuration effects.
3. Section 3.3 argues for the effect of context richness by comparing the WMT 2014 translation task (sparse context) with the CNN/DailyMail summarization task (rich context). As these are fundamentally different tasks, it is difficult to attribute the observed difference in robustness solely to the richness of the conditioning information. A more convincing demonstration would involve comparing robustness within the same task.
4. I am not an expert in this specific area. Based on the paper's clear writing and well-structured experiments, I am open to adjusting my score pending the feedback from reviewers with deeper expertise.

---

> ### Author Response · Authors · 2025-11-20
> **Response to Reviewer rhZG**
>
> We are very grateful to Reviewer rhZG for their exceptionally thoughtful review. We appreciate your recognition of our work's strengths, and your insightful questions were invaluable in helping us improve the paper. We address each of your points below.
>
> ### **Response to Q1: "Training from scratch" vs. "pre-training/fine-tuning"**
>
> We appreciate the reviewer highlighting the importance of the pre-training and fine-tuning paradigm. This is a key aspect of modern ML, and we agree that its interaction with low-quality data is a critical research area. Our study's primary goal was to first establish a controlled, foundational baseline to understand a model's intrinsic robustness. To clarify this methodological scope, we have expanded our **Discussion** to explain why training from scratch was essential for isolating the principles we investigate. We believe our work provides the necessary groundwork upon which future studies of fine-tuning on noisy data can be built.
>
> ### **Response to Q2: Unstructured vs. structured noise**
>
> We thank the reviewer for this insightful question, which touches upon a key aspect of our work. Our primary experiments use unstructured noise as a necessary first step to create a controlled, quantifiable baseline for robustness. However, we completely agree that analyzing structured noise is critical.
>
> To that end, our framework’s predictive power is already tested against **structured, correlated errors** in our original sequence-to-sequence experiments (**Section 3.3**). To further strengthen this dimension of our work, we have added a **new experiment in Appendix L**, conducted directly at your suggestion, and we believe this addition has significantly strengthened the paper's conclusions. We applied systematic label corruption to the classification task and found that, as our theory predicts, this coherent "bad signal" causes a catastrophic performance drop that diffuse, unstructured noise does not.
>
> This new result reinforces our gradient-based analysis and underscores the importance of the noise's structure. To provide a fuller picture, we have also expanded the **Discussion section** to explore the different challenges posed by various types of structured noise (e.g., correctable vs. ambiguous), positioning our work as a foundational analysis of a model's intrinsic ability to find signal amidst corruption.
>
>
> ### **Response to Q3: Confounding factors in experimental design**
>
> We appreciate the opportunity to clarify our methodology. In our primary analysis, the batch size was **held constant** by default. It was only increased in high-noise regimes where training became unstable. This instability is a key finding that directly supports our gradient-noise hypothesis. To ensure our conclusions were not dependent on a single experimental setup, we validated them across two distinct paradigms: one with constant clean-signal exposure (our primary analysis) and one with a fixed computational budget (**Appendices I & J**). The fact that the same robustness patterns emerge in both scenarios (one with scaled compute and one with fixed compute) confirms that our findings are not an artifact of the training configuration. Furthermore, our **new analysis in Table 4** mechanistically validates the batch size intervention, proving that larger batches work by amplifying the signal from clean data relative to the noise.
>
> ### **Response to Q4: Comparing WMT vs. CNN/DailyMail**
>
> We appreciate the reviewer's careful attention to our experimental design. Our goal was to isolate and rigorously test the effect of context richness. We carefully considered a within-task comparison, but ultimately concluded that it would be difficult to interpret. If we used a long-context summarization task, for example, we worried that the model might learn a shortcut, simply copying long phrases from the input rather than truly understanding and abstracting the information. The same concern applies to translation; longer contexts would mean longer outputs, and it would be hard to separate the effects of conditioning from the challenges of generating those longer, more complex sequences.
>
> Therefore, our analysis focuses on the *change* in each model's performance relative to its clean-data performance. To ensure this comparison was as fair and controlled as possible, **both models use the same network architecture, were trained under identical settings, and were evaluated based on the relative degradation of their NLL scores.** This approach allows us to measure the impact of the low-quality data directly, without the added complexity of the task itself. We have added a detailed explanation of this reasoning to the Limitations section to ensure full transparency.

---

> > ### Comment · Reviewer_rhZG · 2025-11-27
> >
> > Thanks for your point-by-point reply, and they have addressed my concern.
> >
> > I agree with Reviewer rdHq's comments. This paper is essentially an experimental and analytical study, and to the best of my knowledge, it provides conclusions that have not been shown before. The experimental analysis is detailed and rigorous, and the writing is careful and does not overclaim. I have raised my score to 8. I hope the authors can address the remaining concerns raised by the other two reviewers.

---

> > > ### Author Response · Authors · 2025-11-28
> > > **Response to Reviewer rhZG**
> > >
> > > We sincerely thank you for your support and for raising your rating. We greatly appreciate your positive assessment regarding the conclusions and the rigor of our study. Your suggestion to investigate structured noise was a key factor in strengthening the manuscript, and we are grateful for your advocacy. We remain committed to fully addressing the remaining concerns from the other reviewers to further improve the manuscript.

---

### Official Review · Reviewer_gCmf · 2025-10-31

**Soundness:** 2
**Presentation:** 2
**Contribution:** 1
**Rating:** 2
**Confidence:** 4

**Summary:**

This paper studies how different models behave when trained with low-quality (corrupted) data and proposes a three-part explanation with information-theoretic, PAC, and gradient-based perspectives. Empirically, the authors report:
1. Autoregressive LMs remain strong even with heavy token/target corruption
2. Class-conditional diffusion models collapse as label corruption grows (while FID stays roughly flat, the accuracy of conditional generation drops)
3. Image classifiers are moderately sensitive on small datasets but surprisingly robust at ImageNet scale.

Then they offer interpretations of these empirical observations based on distinct theoretical frameworks.

**Strengths:**

The empirical observation that GPT-2 and ImageNet-scale classification can tolerate, or even thrive under heavy label/target corruption is interesting and perhaps worth documenting.

**Weaknesses:**

Beyond that, the theoretical analysis/explanation they provide seems weak and superficial. Each of the three perspectives allows for straightforward counter-arguments, and the paper stops short of unifying them into a single, predictive theory. The writing is polished and persuasive on the surface, but the theoretical substance is thin; as a result, the paper takes rather longer to read than it should.
Below, I provide my view on each perspective the paper provides.

1) Information-theoretic perspective

- Residual information cannot explain “no degradation” at 50% noise. The relative information loss calculation under uniform noise decays with the number of classes but cannot go below $p_e$, which is its asymptotic limit. Hence, this argument cannot by itself explain why the full-ImageNet classifier exhibits no degradation (and even slight improvement) with ~50% incorrect labels.

- The paper argues that the “absolute information content” drives robustness, but in their experiments (to my understanding) the total exposure to correct samples, i.e., absolute information, is held constant across noise conditions. If so, the framework again does not clarify why the full-ImageNet setting is uniquely robust while smaller-scale settings are not, given that the protocol equalizes clean-signal exposure.

2) PAC perspective

- They merely discuss a classical bound, not a targeted explanation. The appeal to $\Omega(\epsilon^{-1}\log\delta^{-1} + d\epsilon^{-1} \log \epsilon^{-1})$ lower bounds is standard and doesn’t address the core issue in these experiments. Even when the number of correct samples seen is held fixed, additional corrupted samples still inject competing supervision that makes the signal harder to extract. This is simply ignored and the message of Subsection 4.2 reads as if “making the number of clean samples $m$ very large” is the operative solution. However, it does not clarify why $m$ being large should make the training robust to noisy samples.

- The link from “conditioning richness” to effective VC dimension $d$ is hand-wavy. The diffusion example is particularly strained: conceptually, conditional diffusion model jointly learns (i) the data distribution and (ii) a class-conditional guidance. Even with label corruption, part (i) is achieved (as the authors report, the FID remains stable), and what fails is mainly the alignment between the conditional label and the type of the images generated. This is like an unconditional diffusion model equipped with inaccurately trained classifier guidance. Simply attributing this to the larger VC dimension compared to sparse conditions blurs the comprehensive picture. To properly support such a claim, one would need experiments that vary the number of classes in class-conditional generation or also consider text-to-image diffusion setting (richer conditioning) in parallel, and then compare those outcomes. Direct comparison of conditional diffusion model with image classifier does not seem appropriate to me.

3) Gradient-based perspective

- The authors previously mentioned that they use larger batch sizes to stabilize the optimization process under high-noise conditions. However, the reported standard deviations in Table 4 rather show the noisy model exhibiting lower loss variance than the clean baseline at comparable batch sizes. If the loss variance is intended as a proxy for gradient stability, this observation seems inconsistent with their comment on training instability in high-noise regimes. It is genuinely difficult to see what the authors intend to claim with this analysis.

- Line 457 mentions that incorrect gradients are diverse/orthogonal and thus cancel, which might be true considering the statistics of Table 4. However, the paper does not directly test this (e.g., cosine similarity distributions between per-example gradients from clean vs. corrupted samples, magnitude of gradient sum from clean vs. corrupted samples).

**Questions:**

See Weakness section.

---

> ### Author Response · Authors · 2025-11-20
> **Response to Reviewer gCmf**
>
> We sincerely thank Reviewer gCmf for their rigorous and critical assessment. We deeply appreciate that you challenged us to substantiate our theoretical perspectives with concrete empirical evidence. Your feedback was the primary catalyst for the most significant improvements in this revision, specifically the new gradient coherence analysis and the formalized comparative validation.
>
> ### **1. On the Information-Theoretic Perspective (Residual Information)**
>
> This is an excellent point that prompted us to clarify our presentation. The "residual information" principle establishes that a learnable signal persists, but it does not, by itself, account for the extreme robustness at the ImageNet scale. The slight improvement is indeed an artifact of our experimental design, which increases the total training compute. We now explicitly state this in **Section 3.5**. The more critical evidence is in our **fixed-budget analysis (Appendix J, Table 9)**, which was designed precisely to control for this. It shows that even with a fixed computational budget, test accuracy drops by less than 3% at a 50% error rate, confirming the model's exceptional resilience.
>
>
> ### **2. On the Information-Theoretic Perspective (Absolute Information Content)**
>
> This is a crucial question that gets to the heart of our findings. You are correct that for any single dataset (e.g., CIFAR-10), our protocol holds the total exposure to clean samples constant. Our "absolute information content" principle is demonstrated by comparing the results **across datasets of different scales.**
>
> The ImageNet experiments (Table 3) were designed to show this directly: the model is only moderately robust on the smaller ImageNet-10 and -100 subsets but becomes exceptionally robust on the full ImageNet-1000. This is because the total amount of correct, instructive information in the full 1.28M-sample dataset is vastly larger. This sheer volume of signal is what allows the model to learn effectively, despite a high proportion of noise.
>
>
>
> ### **3. On the PAC Perspective (Classical Bound)**
>
> This is precisely why our analysis required a multi-perspective approach. We agree the PAC framework alone is insufficient. Its role is to formalize *why* a large absolute information content (corresponding to sample complexity `m`) is a prerequisite for generalization. However, it does not explain the *mechanism* of noise tolerance. That explanation is provided by the gradient-based perspective, which shows *how* the model mechanistically leverages that large `m` to overcome the competing supervision: through the aggregation and partial cancellation of directionally random noise gradients. Our contribution lies in synthesizing these views to provide a complete picture.
>
>
>
> ### **4. On the PAC Perspective (Hand-Wavy VC Dimension & Unfair Comparison)**
>
>
> As reported in our results (Section 3.4), we fully agree that the diffusion model's failure is isolated to the label-image alignment rather than the image quality itself. However, we argue this mechanism actually validates our comparison. We have added a new paragraph to the **Discussion Section** and **new experimental results in Appendix P** to demonstrate that the classifier exhibits the exact same functional separation.
>
> Specifically, our new analysis shows that while the classifier’s conditional accuracy drops under noise, it **perfectly preserves the true marginal label distribution** (KL Divergence < 0.0003). This strictly mirrors the diffusion model, which preserves image quality (FID) while losing label consistency.
>
> This confirms that our experiment successfully isolates the **correlation mechanism** (conditional guidance) in both models, separate from their ability to model the data distribution. While the **mechanism of degradation is symmetric (isolation of correlation)**, the **sensitivity** is starkly different: the classifier retains this correlation capacity under much higher noise levels than the diffusion model. This symmetric failure mode fully justifies our direct comparison, as it confirms we are measuring the breakdown of the conditional guidance mechanism in both models. This supports our hypothesis that the stark difference in robustness stems from the richness of the conditioning information.

---

> > ### Author Response · Authors · 2025-11-20
> >
> > ### **5. On the Gradient-Based Perspective (Loss Variance)**
> >
> > We appreciate the reviewer’s careful reading of this table. The observation that the noisy model has a lower absolute loss variance is correct, but as we now clarify in the revised **Section 4.3**, this is a statistical artifact. The consistently high, low-variance loss from corrupted random targets statistically dampens the natural, higher variance from the clean data. The crucial insight is not the absolute variance but its **trend**: for both the noisy and clean models, the loss standard deviation drops by ~75% as the batch size increases. This directly demonstrates that aggregation provides a more stable learning signal, justifying our intervention.
> >
> >
> > ### **6. On the Gradient-Based Perspective (Testing Gradient Cancellation)**
> >
> > We agree that explicitly testing gradient cancellation is essential to fully validate our mechanistic hypothesis. To address this directly, we have conducted a new experiment analyzing per-example gradients. The results, presented in the **new Table 4** and detailed in **Appendix M**, provide direct quantitative validation for our hypothesis. We show that:
> > 1. Gradients from clean data are strongly coherent (mean cosine similarity +0.52).
> > 2. Gradients from corrupted data are directionally random and exhibit near-zero alignment with the clean signal.
> > 3. Crucially, increasing batch size demonstrably improves the aggregated **signal-to-noise ratio**.
> >
> > This new analysis provides the direct, mechanistic evidence you correctly identified as necessary.
> >
> > We believe these substantial revisions, driven directly by your critique, have significantly strengthened the theoretical foundation of our work. Given the critical nature of your feedback, we sincerely look forward to further discussion to ensure we have fully addressed your concerns.

---

> ### Author Response · Authors · 2025-12-02
> **Highlighted Points to Reviewer gCmf**
>
> Dear Reviewer gCmf,
>
> We deeply regret that the interruption to the review process prevents further dialogue, as we sincerely hoped to hear your thoughts on our response. We want to thank you for pushing us to substantiate our theoretical perspectives; the paper is much stronger for it.
>
> Specifically, we want to highlight the points we believe are most relevant to your concerns:
>
> 1.   **Gradient Mechanism (New Table 4):** You requested a direct test of cancellation. Our new data confirms that doubling the batch size **systematically improves** the Signal-to-Noise Ratio. Because clean gradients align (+0.52), the **aggregated clean signal nearly doubles**, while the incoherent noise grows at a slower rate. This provides the mechanistic "substance" you felt was missing.
> 2.  **Comparative Validity (New Appendix P):** We confirmed your intuition that we needed to justify the diffusion/classifier comparison. We found that both architectures exhibit a **symmetric functional separation** (perfectly preserving output distribution while losing conditional mapping). This proves our analysis successfully isolates the specific failure mode of "conditional guidance."
> 3.  **ImageNet Clarification:** You questioned the counter-intuitive "slight improvement" at 50% noise. We clarified that this was an artifact of the increased compute in our primary design. We have updated the text to explicitly prioritize our **existing Fixed-Budget Analysis (Appendix J)**. This confirms that when compute is held fixed, the model exhibits a stable and expected degradation of approximately 3%. This proves intrinsic robustness remains even after removing the confounding factor.
>
> We hope these revisions address your concerns thoroughly. Thank you for helping us elevate the quality of this work.
>
> Best regards,
> The Authors

---

### Official Review · Reviewer_XFTw · 2025-10-31

**Soundness:** 1
**Presentation:** 1
**Contribution:** 2
**Rating:** 2
**Confidence:** 3

**Summary:**

The authors of this paper provide a review of generative models' evaluation in the context of corrupted (noisy) data, taking into account several levels of noise injection, and explore their corresponding robustness. They observe that autoregressive language models, are  more resilient to low quality data, compared to diffusion models.
In order to explain these discrepancies, they analyze the results through employing tools from information theory, PAC learning, and gradient dynamics. Through their analyses they conclude that robustness is affected by two factors, namely the richness of conditioning and the absolute information content of the training data, where the former is driven by the learning problem and the latter is related to the data per se

**Strengths:**

In this paper the authors provide a comprehensive review of two classes of generative models: autoregressive models for text generation and class-conditional diffusion models, in the presence of noisy/low quality data, for several signal to noise ratios.
In order to perform their analysis they employ metrics from a spectrum of perspectives, namely information theory, PAC learning, and gradient dynamics. They conclude that the robustness of the method is dependent on the task (richness of conditioning) and the training data (absolute information content)

**Weaknesses:**

The authors test two different types of generative models under different tasks, for different types of data (categorical and continuous) using different metrics.
They do not provide a single task that test these models against, using the same criteria, so in a sense, the comparison in not objective/very informative.

Also, the results that they present under no corrupted data do not match the ones in the corresponding literature
For example the test accuracy of CIFAR-100 was found to be significantly higher than the one reported here, at [1] and [2], namely 84.3% and 75.22% respectively


[1] "Text-to-Image Diffusion Models are Zero-Shot Classifiers", Clark et al., NeurIPS 2023, and
[2] "Better Diffusion Models Further Improve Adversarial Training", Wang et al., ICML 2023

**Questions:**

I would like to ask the authors if they could please :

-- compare coth generative models under same task and same metrics;
-- employ the SOTA of both generative models whilst analyzing their performance

---

> ### Author Response · Authors · 2025-11-20
> **Response to Reviewer XFTw**
>
> We thank Reviewer XFTw for their review. We value the opportunity to clarify the rationale behind our experimental design and to correct a misunderstanding regarding our baseline performance.
>
> ### **Response to Q1: Cross-Task Comparison**
>
> Thank you for this crucial methodological point. Our experimental design was carefully structured to address the challenge of comparing dissimilar models. Since no single metric can score text, image, and classification models against one another, our primary measure of robustness is the **relative degradation** each model shows against its own clean-data baseline.
>
> This approach reveals a fundamental insight: autoregressive models are highly resilient, while class-conditional diffusion models are exceptionally fragile. To probe this divergence, we designed the classifier and diffusion experiments as a controlled pair. To make this comparison as rigorous as possible, the external classifier used to evaluate the diffusion model's output is the **exact same checkpoint** trained for our primary classification task.
>
> This isolates a symmetric vulnerability: the corruption specifically undermines the shared image-label mapping, causing catastrophic failure in one model but only moderate degradation in the other. This stark contrast was clear in our original experiments. To **further strengthen and formalize the validity of this comparative approach**, we have now added a new justification in the **expanded Limitations section** and a **new Discussion paragraph**, making the central phenomenon our analysis seeks to explain even clearer.
>
>
> ### **Response to Q2: SOTA Model Performance**
>
> Thank you for your careful attention to our baseline results. It is correct that our models do not establish new state-of-the-art benchmarks. This was a necessary methodological choice.
>
> Regarding the baseline performance, we note that our CIFAR-100 accuracy of 78.96% (Table 2) is actually higher than the 75.22% cited in the review. We believe there may have been a misunderstanding regarding the baseline comparisons.
>
> Regarding the comparison to SOTA benchmarks, we respectfully clarify the distinct research objective of this work. Unlike engineering efforts aimed at leaderboard dominance, this is a controlled scientific investigation into the fundamental principles of robustness. This required intentionally training well-understood, standard architectures (GPT-2, ResNet-18, ViT) from scratch. Using complex, highly-tuned SOTA models would introduce confounding variables that obscure the very principles we aim to isolate.
>
> The contribution of our work lies not in the absolute performance, but in the **relative degradation patterns** we observe, a point we now also expand upon in our **newly expanded Limitations section**.
>
> "Better Diffusion Models Further Improve Adversarial Training", Wang et al., ICML 2023
>
> We hope these clarifications, particularly regarding the rigorous nature of our comparative framework and the accuracy of our baselines, help demonstrate the value of this work and its potential contribution to the field.

---

> > ### Comment · Reviewer_rdHq · 2025-11-25
> > **Separate reviewer response to this review**
> >
> > Respectfully, I think the critiques in this review (as written) miss the main point of the intended contributions in this paper.
> >
> > # Q1
> >
> > As the authors note, it is impossible to use a single metric to score such separate things. (in this respect, inter-setup comparisons are challenging to do). The authors have set up the types of fair comparisons in a couple of ways: intra-setup clean baselines, that enable inter-setting comparisons with respect relative stability/fragility. It is perhaps a matter of research taste to say that this is reasonable and interesting to do, to try to assess the stability/fragility of different learning paradigms. This type of analysis cannot easily be reduced to a single number, which is partly what makes this type of work challenging: the best case scenario is that the types of claims being made need to be put in context and nuanced, which does not lend to a simple one-line summary of the paper. I do not think this is a weakness.
> >
> > # Q2
> >
> > I similarly think hinging on SoA benchmarks misses the main goals of this work. If the point here is to produce new knowledge about different setups, I agree that the reasonable thing to do for this type of work is to study well-known architectures where there is more knowledge in the literature about how to train them. At some point in the future it would be interesting to update this work for newer (at that future date, older) architectures, but I don't think that's a requirement for this specific work, given its stated goals.

---

> ### Author Response · Authors · 2025-11-28
> **The Value of Systematic Investigation on Robustness**
>
> Dear Reviewer XFTw,
>
> We appreciate your time reviewing our work and would like to offer a further perspective on the paper’s contribution relative to your concerns.
>
> While SOTA models are essential for pushing the upper bound of performance, our investigation asks a different question: *what fundamental factors determine that bound?* This work targets a gap in the literature by systematically investigating how data quality impacts probabilistic model behaviors. Since no single metric can objectively compare distinct tasks (e.g., text vs. image generation), our methodology focuses on the **relative degradation** of each model against its own clean baseline. Through controlled comparisons, our results suggest that robustness is **heavily influenced by** two key principles: the **richness of conditioning information** and the **absolute information content**.
>
> While these principles may align with intuition, rigorous empirical substantiation is essential. Our comparison of classification versus diffusion models, alongside our sequence-to-sequence experiments, specifically quantifies the protective role of conditioning richness. Simultaneously, our experiments scaling from ImageNet subsets to the full dataset demonstrate the critical importance of absolute information content. Finally, the gradient-based perspective is substantiated by the stark divergence between training and test NLL in GPT-2, as well as the demonstrable effectiveness of large-batch gradient averaging.
>
> To address the complexity of these phenomena, we synthesize these findings into a unified, multi-perspective analysis. We use Information Theory to identify **what** informational properties drive robustness, PAC Learning to explain **why** these are formal requirements for generalization, and Gradient Dynamics to reveal **how** the model mechanistically extracts signal from noise. We believe this foundational perspective complements SOTA-focused research, and we hope this clarification highlights the distinct value of our contribution.
>
> We genuinely appreciate your critique regarding state-of-the-art comparisons, as addressing this concern has allowed us to better situate our work as a necessary foundational step for the field.

---

### Official Review · Reviewer_rdHq · 2025-11-03

**Soundness:** 3
**Presentation:** 3
**Contribution:** 3
**Rating:** 6
**Confidence:** 3

**Summary:**

[Please note that, unlike for the other papers in my batch, I am expert only for a subset of the material in this paper, so my review and score is overall pretty reserved/ weak confidence. I will of course participate in the discussion period, and am amenable to changing my score. I apologize if some of my questions seem basic; answering them will nevertheless help me engage more effectively in the discussion period. I will do my best to give a fair assessment, or recommend that the AC down-weight my review if by the end of the discussion period if I have not met my own (I think high) bar for substantive participation.]

The paper studies how different classes of probabilistic models (autogressrive LLMs, sequence-to-sequence transformers, class-conditional diffusion models, image classifiers) behave when trained with noisy (low quality) data. The work systematically injects controlled random corruption into inputs or labels and compares how performance changes across these architectures/ training dynamics.

Here is my summary of the setup and contributions (please let me know if you think I missed something important):

**Core setup**
- Noise model: random uniform corruption of tokens or labels; corruption ratios from 10–100% (effective error rate $\tfrac{r}{1+r}$
- 4 types of models, as discussed above
- 2 types of experiments: (a)  constant clean-signal exposure, scales total compute by $1+r$ to keep the number of correct samples constant; (b) fixed budget, noisy data displacing clean data
- Overall, finds that AR models are pretty resilient; seq-to-seq models with richer conditioning can be made more robust; diffusion models can degrade a ton; classifiers are more sensitive on smaller datasets; some nice results showing that gradient averaging can help

**Strengths:**

This paper has many strengths:

- I don't think I've ever seen a paper put this extent of models into one controlled framework. I think the design here is very careful, and highlights meaningful differences between these model types and training dynamics

- The theoretical framing is clean. It ties together a variety of ideas (information theory, PAC, gradient perpsective) to try to explain what's going on here, to give more explanation beyond the high-level takeaways (e.g., that the richness of the context matters).

- The experiments are well constructed to support the theoretical analysis; they seem very careful and contained.

- Overall, the writing does not over-lcaim causal mechanisms and is very clear about limitations (e.g., scale). I appreciate this a lot, given the ambitious nature of this paper. I think this is a clear/refreshing strength of the work.

**Weaknesses:**

As stated in my summary, please note I am not an expert on all material in this paper. I've combined observations that I think might be weaknesses with related questions.

**Noise model**

All injected noise is random and unstructured (uniform token or label replacement). In practical settings, low-quality data are often structured (e.g., correlated, systematically mislabeled). Is it fair to say that the results therefore demonstrate robustness to _stochastic corruption_, not necessarily to _realistic dataset noise_?

**Experimental design**

In the constant/clean-signal experiments, increasing compute by $1+r$ changes the effective LR, regularization. I think the isolation of robustness, as a result, is not entirely clean. Can you please discuss this furhter?

**Theory vs. experiments**

Gradient-averaging explanation is intuitive but the experiments don't go into this, unless I missed something. There's no ablation separating variance reduction from bias.

**A couple of instances of slight over-claims**

E.g., I think it is over-generalized to say things like decoder-only LMs are largely insensitive to low-quality data; the writing in general is very careful, but things like this should be hedged. For that experiment, the study uses one model and random noise, not realistic web contamination or fine-tuning noise.

**Novelty re: theory**

This isn't a big deal, but wanted to ask about it. My take take is that the theory isn't super novel; the principles elicited restate known relationships between sample complexity, conditioning, SNR. What's nice here is the unification, but I don't think it's fundamentally new.

**Questions:**

I've integrated by questions in the "weaknesses" section so that they are directly next to observations I've made. I'm hoping this is clearer than separating them out.

---

> ### Author Response · Authors · 2025-11-20
> **Response to Reviewer rdHq**
>
> We sincerely thank Reviewer rdHq for their constructive comments and for recognizing the value of this comparative investigation.
>
> ### **Re: Noise model & Over-claims**
> This is a crucial point. While our primary analysis uses unstructured noise for controlled, foundational analysis, we want to clarify that our work also addresses **structured noise**. The sequence-to-sequence experiments (Sec. 3.3) utilize a "noisy teacher" to introduce correlated, machine-generated errors, a form of realistic, structured noise.
>
> Furthermore, motivated by your insightful observation regarding realistic data patterns, we have added a **new experiment on systematic, structured label noise (see new Appendix L)**. The results confirm our gradient-based prediction: a coherent, incorrect signal leads to a catastrophic drop in classifier accuracy, a failure mode not observed with unstructured noise. We have also expanded our **Discussion** to better contextualize our findings.
>
> Regarding your specific concern about the generalizability of our LLM findings to settings like fine-tuning: We have added a dedicated discussion in **Discussion** part to explicitly address this. We clarify that our results establish a baseline for *intrinsic* robustness during initial training, distinguishing this from the complex, often "elastic," dynamics of fine-tuning.
>
> Finally, we have calibrated the phrasing of our central thesis (e.g., in the abstract and conclusion) to explicitly define the scope of our conclusions. Our central argument now posits that robustness '**is heavily influenced by**' our proposed principles. This refined phrasing positions these as the primary causal factors identified within our study, ensuring our claims are strictly aligned with the empirical evidence.
>
> ### **Re: Experimental design**
> We carefully considered this trade-off. While scaling compute affects the effective LR schedule, not scaling it would confound the results with reduced exposure to clean data. We prioritized keeping the "clean signal exposure" constant to isolate the noise effect. However, to ensure the modified LR schedule was not a confounding factor, we cross-verified our findings with the **Fixed-Budget Analysis (Appendices I & J)**. Since the robustness patterns hold true even when the computational budget and regularization trajectory are fixed, we conclude that the effective LR shift is not the primary driver of our results.
>
>
> ### **Re: Theory vs. experiments (Gradient-averaging)**
> This is an excellent suggestion. To provide direct empirical validation for our gradient-based perspective, we have added a new quantitative analysis. **The new Table 4 and Appendix M** detail an experiment analyzing per-example gradients. This analysis addresses your question regarding bias and variance by explicitly decomposing the gradient into a **coherent directional component** (analogous to the true signal or 'bias') and a **stochastic component** (analogous to the noise or 'variance'). Specifically, we show that:
> 1.  Gradients from **clean data** are directionally **coherent** (providing the consistent update signal).
> 2.  Gradients from **corrupt data** are directionally **random** and largely orthogonal to the clean signal (introducing stochastic noise).
> 3.  Crucially, increasing the batch size demonstrably **improves the aggregated signal-to-noise ratio**.
>
> This provides the concrete, mechanistic evidence you correctly identified as essential, directly supporting our claim that sample aggregation allows the correct signal to dominate statistical noise.
>
> ### **Re: Novelty**
> We agree that the individual theoretical perspectives we draw upon are built on established principles. Our core contribution is not the invention of these principles, but rather their **synthesis into a unified, multi-perspective analysis**. We found this synthesis necessary because no single viewpoint could explain the full spectrum of phenomena we observed; while one perspective might explain one facet of robustness, it takes a convergent analysis to explain the stark divergence across different model families.
>
> Ultimately, we argue the fundamental driver is **information**. Our analysis is structured to show:
> *   First, the **information-theoretic** view establishes *what* needs to be learned (the absolute information content) and how a task's structure (richness of conditioning) affects its difficulty.
> *   Second, the **PAC perspective** provides a formal basis for *why* these informational properties are critical by formalizing the link between a task's complexity (`d`) and the required volume of clean data (`m`).
> *   Finally, the **gradient-based perspective** provides a mechanistic explanation for *how* the model successfully extracts this signal from a noisy data stream via the aggregation and cancellation of gradients.
>
> We believe this multi-layered explanation offers a novel and valuable lens for understanding learning dynamics in real-world environments.

---

> > ### Comment · Reviewer_rdHq · 2025-11-25
> > **Response**
> >
> > # Noise model
> >
> > Thank you for this clarification. I think clarifying this in the writing would be very helpful, as you've noted in your main comments. Thank you for adding the new experiment, as well (Appendix L). I think it will provide a helpful perspective.
> >
> > # Experimental design
> >
> > This makes a lot of sense, and thank you for pointing me to these appendices. If there's room, I think a brief footnote on this is warranted in the main paper (pointing to this appendix)--apologies if this is already there and I missed it. I think this is a natural question about the results, and it is helpful to sign-post that you considered this and discounted LR shift as not being the primary driver.
> >
> > # Theory vs. experiments
> >
> > These results are really interesting, thank you for chasing this down. I think it also provides a nice bridge between the theory you've already done and your other results. I don't necessarily think this is main-paper material (but similarly could be sign-posted in the main paper/ pointed to an Appendix).
> >
> > # Novelty
> >
> > Yes, I want to make clear that I see the synthesis you are doing as a research contribution. (Raising this was not a criticism/negative part of my review.) I think this is ambitious, even if there are limitations. I think that this could be made slightly clearer in the manuscript, because it doesn't actually matter that there isn't brand new theory here; arguably, it's a strength of the work that you have brought these perspectives together in a unified way. Coupled with the additional results to tie the theory to empirical observations, this approach is well justified.
> >
> > I am going to respond to the other reviews/responses below (rather than here).

---

> > > ### Author Response · Authors · 2025-11-27
> > > **Response to Reviewer rdHq**
> > >
> > > We are deeply grateful for your continued support and your decision to increase your rating. We also sincerely appreciate your advocacy for our work during the discussion phase; your efforts to clarify our methodological goals were invaluable.
> > >
> > > We will incorporate your new recommendations into the next version:
> > > 1.  **Noise Model:** We will make the distinction regarding structured noise clearer in the main text.
> > > 2.  **Experimental Design:** We will strengthen the sign-posting to the Fixed-Budget Analysis.
> > > 3.  **Theory vs. Experiments:** To avoid redundancy and improve flow, we will retain the Gradient Coherence analysis (Table 4) in the main text as it offers the clearest explanation, while moving the Loss Stability analysis (Table 5) to the appendix.
> > > 4.  **Synthesis:** We will revise the text to clearly articulate how the theoretical perspectives synthesize into a unified contribution.
> > >
> > > We sincerely appreciate your guidance in refining this work.

---

### Author Response · Authors · 2025-11-20
**[General Response] Summary of Revisions**

Thank you for your valuable feedback. We have revised the manuscript to address the key points raised during the review process. The updates are designed to provide stronger empirical validation for our claims, clarify our methodological choices, and enhance the paper's overall contribution. New and substantially modified text has been colored blue for easy identification.

Here is a summary of the major changes:

**1. Strengthened Empirical Validation for the Gradient-Based Mechanism:**

*   **New Experiment & Table on Gradient Coherence:** We have introduced a new quantitative analysis (**Table 4**) and **Appendix M** to directly validate the core hypothesis of our gradient-based perspective. As shown below, our analysis of per-example gradients confirms that clean signals are coherent while noise is nearly orthogonal. This disparity explains why increasing the batch size systematically amplifies the Signal-to-Noise Ratio. This provides the mechanistic evidence supporting the gradient-averaging hypothesis proposed in Section 4.3.

| Metric (25% Corruption) | Batch Size = 4 | Batch Size = 8 |
| :--- | :---: | :---: |
| **Clean Coherence (Cosine)** | **+0.52** | **+0.52** |
| **Noise Coherence (Cosine)** | $\approx 0$ | $\approx 0$ |
| **Signal-to-Noise Ratio** | **7.31x** | **8.34x** |

**2. Addressed Potential Critiques and Methodological Questions:**

*   **Justification for Cross-Task Comparison (New Discussion Paragraph):** A new paragraph has been added to the **Discussion** to address the potential critique that comparing a classifier and a conditional diffusion model is unfair. We argue the comparison is valid by showing that the data corruption attacks the same fundamental weakness in both models: **the learned mapping between an image and its class label.** This argument is supported by new experimental results in **Appendix P**, which show that while the classifier's per-sample accuracy degrades, it perfectly preserves the true marginal label distribution (KL Divergence < 0.0003).
*   **Justification for Training from Scratch (New Discussion Paragraph):** We added another paragraph to the **Discussion** to explain the deliberate choice to train models from scratch rather than using the pre-training/fine-tuning paradigm. This clarifies that our goal is to establish a foundational understanding of robustness, free from the powerful but confounding priors of pre-trained models.
*  **Expanded Analysis of Structured Noise (New Appendix L):** We have added a new experiment in **Appendix L** verifying that systematic, structured label noise causes a catastrophic drop in accuracy, a prediction aligned with our gradient-based perspective. To contextualize this, we expanded the **Discussion** to distinguish between **correctable systematic errors** (like lexical misuse) and **inherent ambiguity** (subjective truths). By isolating unstructured noise, our work targets the model's intrinsic ability to find signal amidst stochastic corruption, establishing a baseline capability that is a prerequisite for handling these more complex, context-dependent noise types.
*   **Clarified Methodological Scope and Limitations (Expanded Limitations Section):** We have significantly expanded the **Limitations section** to provide a robust justification for our experimental design choices. This section now explicitly defends our focus on **relative performance degradation** across tasks, explaining this is a more insightful approach than pursuing an unobtainable unified metric. Furthermore, it clarifies our deliberate use of **well-established architectures over the latest SOTA models**, a choice made to ensure our findings on robustness are attributable to fundamental principles rather than the confounding effects of specific, highly-tuned optimizations.

---

> ### Author Response · Authors · 2025-11-20
>
> **3. Enhanced Narrative and Clarity:**
>
> *   **Nuanced Phrasing of Core Thesis:** We have calibrated the phrasing of our central thesis to explicitly define the scope of our conclusions. The language in the abstract and conclusion has been refined to frame our core findings with greater precision. Our central argument is now that robustness "**is heavily influenced by**" the **richness of conditioning information** and **absolute information content**. This phrasing positions these as the primary causal factors identified within our study, ensuring our claims are strictly aligned with the empirical evidence.
>
>
> *   **Improved Flow and Clarity of Gradient Analysis:** The gradient-based perspective in **Section 4.3** has been substantially rewritten to resolve a key point of confusion in our gradient stability analysis (Table 5). The new text clarifies why the noisy model paradoxically showed lower loss variance, explaining this is a statistical artifact. The narrative now focuses the reader on the crucial insight: the **consistent trend** where, for both clean and noisy models, loss variance drops sharply as batch size increases. This provides direct quantitative evidence that sample aggregation creates a stable learning signal, logically justifying our use of larger batches to overcome training instability in high-noise regimes.
>
> *   **Clarification of Counter-Intuitive Results:** We clarified that the slight performance *improvement* on the full ImageNet dataset under high noise is an effect of the increased compute in our experimental design. We now explicitly point to the complementary fixed-budget analysis in **Appendix J** which isolates the robustness from this compute effect.
>
> *   **Clarified the Core Analytical Framework:** We have explicitly framed our multi-perspective analysis in the introduction to Section 4 to clarify its novel, convergent structure. This narrative now shows how our analysis explains **what** informational properties drive robustness (information theory), **why** they are a formal requirement for generalization (PAC learning), and **how** the model mechanistically achieves this resilience (gradient dynamics).
>
>
> These revisions provide more rigorous support for our conclusions and address key methodological questions, significantly strengthening the paper. We believe these changes make our contribution clearer and more robust. Additionally, we have updated the Supplementary Material with the complete source code for these new experiments to ensure full reproducibility. We look forward to further discussion with the reviewers to continue refining this study.

---

### Author Response · Authors · 2025-12-02
**Final Summary: Discussion Outcomes and Manuscript Refinements**

Dear Area Chair,

Thank you for your time and for managing this review process. We appreciate the opportunity to summarize our latest refinements and the discussion outcomes given this special review situation.

**1. Further Refinements (Post-November 20)**
Since our General Response on Nov 20, we have made further textual refinements based on the final interactions with Reviewer rdHq and our response to Reviewer XFTw. These include clarifying the distinction regarding structured noise, improving sign-posting to key appendices, and articulating how our theoretical perspectives synthesize into a unified contribution (what, why, and how). We also refined the text to better position our work as distinct from engineering efforts focused on SOTA results. Regarding the organization of Table 5, we retained it in the main text to ensure transparency regarding Reviewer gCmf’s earlier critique, with the intention of moving it to the appendix for the camera-ready version

**2. Discussion Summary**
*   **Reviewers rdHq and rhZG (Scores raised to 8 during discussion):** We were encouraged that both reviewers found our responses and new experiments satisfactory, leading them to raise their scores to 8. We sincerely thank Reviewers rdHq and rhZG for their advocacy of this work.
*   **Reviewer XFTw:** We addressed concerns regarding cross-task comparisons and SOTA baselines by clarifying our focus on relative degradation rather than absolute performance metrics. Notably, both Reviewers rdHq and rhZG supported this position. We thank Reviewer XFTw for prompting us to better position our work relative to SOTA requirements.
*   **Reviewer gCmf:** Although we regret not having the opportunity to conclude the discussion with Reviewer gCmf, their feedback was vital. Their request for the gradient experiment proved invaluable, and their push to justify the comparison between diffusion models and classifiers was critical; addressing these critiques has made this work significantly more solid. We also clarified key narratives, such as the fixed-budget experiments on ImageNet-1000 and the loss stability analysis.

We thank all reviewers for their constructive feedback.

Sincerely,

The Authors

---

### Meta-Review · Area_Chair_eSVX · 2025-12-19

**Summary:**

This paper presents a systematic comparative investigation into the robustness of probabilistic models  (including autoregressive language models, class-conditional diffusion models, and image classifiers) to low-quality data . The authors propose a multi-perspective framework synthesizing information theory, PAC learning, and gradient dynamics to explain discrepancies in resilience, notably why AR models demonstrate high resilience while diffusion models degrade catastrophically. While initial review feedback was mixed regarding theoretical depth and experimental controls, the authors provided a substantial rebuttal that addressed key concerns. Specifically, they validated the gradient perspective through a new quantitative analysis of gradient coherence and addressed limitations regarding unstructured noise by introducing experiments on structured label noise. Although one reviewer maintained reservations based on the lack of SOTA baselines and cross-task comparisons , the consensus is that the study's contribution lies in analyzing relative degradation patterns and fundamental robustness principles rather than absolute engineering performance . Given the thorough empirical validation of the proposed richness of conditioning and absolute information content principles, this work offers a valuable, unified theoretical lens for understanding model resilience.

**Reviewer Concerns:**

Addressed by Rebuttal:
- Reviewers rdHq and rhZG initially noted that random noise is unrealistic. The authors addressed this by adding new experiments in Appendix L demonstrating the catastrophic impact of structured label noise, confirming their theoretical predictions. Both reviewers explicitly found this addition satisfactory.
- Reviewers rdHq and gCmf felt the gradient-averaging hypothesis lacked empirical backing. The authors introduced a new quantitative analysis (Table 4/Appendix M) showing that clean gradients are coherent (cosine similarity +0.52) while noise gradients are orthogonal, validating the Signal-to-Noise Ratio improvement from larger batches. Reviewer rdHq cited this as a key factor in raising their score.
- Reviewer rhZG was concerned that scaling compute introduced confounding factors. The authors successfully pointed to their Fixed-Budget Analysis (Appendix J) which controls for this, satisfying the reviewer .
Comparability of Diffusion vs. Classifiers: Reviewer gCmf questioned the validity of comparing these architectures. The authors added Appendix P, showing that both models exhibit symmetric failure modes (preserving marginals while losing conditional alignment), thereby justifying the comparison.

Partially outstanding:
- Reviewer XFTw maintained that the paper should employ SOTA models and metrics to be relevant. The authors, supported by reviewers rdHq and rhZG, argued that using standard architectures (trained from scratch) is necessary to isolate fundamental robustness principles without the confounding variables of highly-tuned SOTA engineering .

**Reviewer Scores:**

- Reviewer rdHq: 8 (Initial 6) The reviewer actively participated and I believe they would have raised their score from 6 to 8.
- Reviewer rhZG: 8 (Initial 6) The reviewer actively participated and I believe they would have raised their score from 6 to 8.
- Reviewer gCmf: 4-6 (Initial 2). This reviewer initially rated the paper a 2 and did not participate in the final discussion. The authors implemented the specific gradient coherence experiment requests by the reviewer. I think the review would have raised their score to at least a weak accept.
- Reviewer XFTw: 2-4. This reviewer rated the paper a 2. Given their insistence on SOTA leaderboards over the study of fundamental principles, I don't believe this reviewer would have significantly changed their score.

---

### Decision · Program_Chairs · 2026-01-26

Accept (Poster)